# Bundle Neural Networks
## for message diffusion on graphs

**Jacob Bamberger**[1*] **Federico Barbero**[1] **Xiaowen Dong**[1] **Michael Bronstein**[1,2]
[1]University of Oxford  [2]AITHYRA

## Abstract

The dominant paradigm for learning on graphs is message passing. Despite being a strong inductive bias, the local message passing mechanism faces challenges such as over-smoothing, over-squashing, and limited expressivity. To address these issues, we introduce Bundle Neural Networks (BuNNs), a novel graph neural network architecture that operates via *message diffusion* on *flat vector bundles* — geometrically inspired structures that assign to each node a vector space and an orthogonal map. A BuNN layer evolves node features through a diffusion-type partial differential equation, where its discrete form acts as a special case of the recently introduced Sheaf Neural Network (SNN), effectively alleviating over-smoothing. The continuous nature of message diffusion enables BuNNs to operate at larger scales, reducing over-squashing. We establish the universality of BuNNs in approximating feature transformations on infinite families of graphs with injective positional encodings, marking the first positive uniform expressivity result of its kind. We support our claims with formal analysis and synthetic experiments. Empirically, BuNNs perform strongly on heterophilic and long-range tasks, which demonstrates their robustness on a diverse range of challenging real-world tasks.

## 1 Introduction

Graph Neural Networks (GNNs) (Sperduti, 1993; Scarselli et al., 2009; Defferrard et al., 2016) are widely adopted machine learning models designed to operate over graph structures, with successes in diverse applications such as drug discovery (Stokes et al., 2020), traffic forecasting (Derrow-Pinion et al., 2021), and recommender systems (Fan et al., 2019). Most GNNs are *Message Passing Neural Networks* (MPNNs) (Gilmer et al., 2017), where nodes exchange messages with immediate neighbours. While effective, MPNNs face critical challenges such as over-smoothing (Li et al., 2018; Oono & Suzuki, 2020; Cai & Wang, 2020), over-squashing (Alon & Yahav, 2021; Topping et al., 2022; Di Giovanni et al., 2023), and limited expressivity (Xu et al., 2019b; Morris et al., 2019).

Over-smoothing occurs when node features become indistinguishable as the depth of the MPNN increases, a problem linked to the stable states of the heat equation on graphs (Cai & Wang, 2020). While Sheaf Neural Networks (Bodnar et al., 2022) address this by enriching the graph with a *sheaf* structure that assigns linear maps to edges and results in richer stable states of the corresponding heat equation, they remain MPNNs and inherit other limitations such as over-squashing, which restricts the amount of information that can be transmitted between distant nodes.

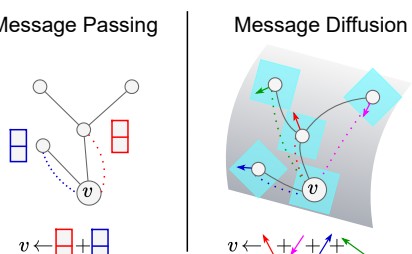

Figure 1: Local message passing on graphs versus global message diffusion on bundles.

We propose Bundle Neural Networks (BuNNs), a new type of global GNN that operates over *flat vector bundles* – structures analogous to connections on flat Riemannian manifolds that augment the graph by assigning to each node a vector space and an orthogonal map. BuNNs *do not perform 'explicit' message passing* through multiple steps of information exchange between neighboring nodes, but instead operate via **message diffusion**. Each layer involves a node update step, and a

---

*jacob.bamberger@cs.ox.ac.uk

diffusion step evolving the features according to a vector diffusion PDE as in Singer & Wu (2012). The resulting architecture enjoys the desirable properties of Sheaf Neural Networks, in that they can avoid over-smoothing, but are global models that can operate at larger scales of the graph to tackle over-squashing, and it achieves better performance on a range of benchmark datasets. Additionally, we prove that equipped with injective positional encodings, BuNNs are **compact uniform approximators**, a new type of universality result for feature transformation approximation.

In summary, our **contributions** are the following:

- We derive BuNNs from heat equations over flat vector bundles, and show that flat vector bundles are more amenable to computation than general vector bundles (Section 3).
- We prove that the diffusion process can mitigate over-smoothing and over-squashing (Section 4), and support these claims with novel synthetic experiments (Section 6.1).
- We prove that, with injective positional encodings, BuNNs are compact uniform universal approximators. To the best of our knowledge, this is the first of such results (Section 5).
- We show that BuNNs perform well on heterophilic and long-range tasks, for instance, achieving a new state-of-the-art result on the `Peptides-func` dataset (Section 6.2).

## 2 BACKGROUND

**Graphs**. Let $G = (V, E)$ be an undirected graph on $n = |V|$ nodes with edges $E$. We represent the edges via an adjacency matrix $\mathbf{A} \in \mathbb{R}^{n \times n}$ where the entry $\mathbf{A}_{uv}$ for $u, v \in V$ is 1 if the edge $(u, v) \in E$ and 0 otherwise. Let $\mathbf{D}$ be the diagonal degree matrix with entry $\mathbf{D}_{vv} = d_v$ equal to the degree of $v$. The graph Laplacian is defined as $\mathbf{L} := \mathbf{D} - \mathbf{A}$ and the random walk normalized graph Laplacian is defined as $\mathcal{L} := \mathbf{I} - \mathbf{D}^{-1}\mathbf{A}$. We assume that at each node $v \in V$ we are given a $c$-dimensional signal (or node feature) $\mathbf{x}_v \in \mathbb{R}^c$ and group such signals into a matrix $\mathbf{X} \in \mathbb{R}^{n \times c}$.

**GNNs and feature transformations**. A *feature transformation* on a graph $G$ is a permutation-equivariant map $f_G : \mathbb{R}^{n \times c_1} \to \mathbb{R}^{n \times c_2}$ that transforms the node signals. A $\text{GNN}_\Theta$ is a (continuous) map parameterized by $\Theta$ that takes as input a graph alongside node signals $G = (V, E, \mathbf{X})$ and outputs a transformed signal $(V, E, \mathbf{X}')$. A GNN on a graph $G$ is therefore a feature transformation $\text{GNN}_\Theta : \mathbb{R}^{n \times c_1} \to \mathbb{R}^{n \times c_2}$. Given a collection of graphs $\mathcal{G}$, a feature transformation $F$ on $\mathcal{G}$ is an assignment of every graph $G \in \mathcal{G}$ to a feature transformation $F_G : \mathbb{R}^{n_G \times c_1} \to \mathbb{R}^{n_G \times c_2}$. The set of continuous feature transformations over a collection of graphs in $\mathcal{G}$ is denoted $\mathcal{C}(\mathcal{G}, \mathbb{R}^{c_1}, \mathbb{R}^{c_2})$.

**Cellular sheaves**. A *cellular sheaf* (Curry, 2014) $(\mathcal{F}, G)$ over an undirected graph $G = (V, E)$ augments $G$ by attaching to each node $v$ and edge $e$ a vector space space called *stalks* and denoted by $\mathcal{F}(v)$ and $\mathcal{F}(e)$, usually the stalks are copies of $\mathbb{R}^d$ for some $d$. Additionally, every incident node-edge pair $v \trianglelefteq e$ gets assigned a linear map between stalks called *restriction maps* and denoted $\mathcal{F}_{v \trianglelefteq e} : \mathcal{F}(v) \to \mathcal{F}(e)$. Given two nodes $v$ and $u$ connected by an edge $(v, u)$, we can *transport* a vector $\mathbf{x}_v \in \mathcal{F}(v)$ from $v$ to $u$ by first mapping it to the stalk at $e = (v, u)$ using $\mathcal{F}_{v \trianglelefteq e}$, and mapping it to $\mathcal{F}(u)$ using the transpose $\mathcal{F}_{u \trianglelefteq e}^T$. As a generalization of the graph adjacency matrix, the *sheaf adjacency matrix* $\mathbf{A}_\mathcal{F} \in \mathbb{R}^{nd \times nd}$ is defined as a block matrix in which each $d \times d$ block $(\mathbf{A}_\mathcal{F})_{uv}$ is $\mathcal{F}_{u \trianglelefteq e}^T \mathcal{F}_{v \trianglelefteq e}$ if there is an edge between $u$ and $v$ and $\mathbf{0}_{d \times d}$ otherwise. Similary, we define the block diagonal *degree matrix* $\mathbf{D}_\mathcal{F} \in \mathbb{R}^{nd \times nd}$ as $(\mathbf{D}_\mathcal{F})_{vv} := d_v \mathbf{I}_{d \times d}$, and the sheaf Laplacian is $\mathbf{L}_\mathcal{F} := \mathbf{D}_\mathcal{F} - \mathbf{A}_\mathcal{F}$. These matrices act as bundle generalizations of their well-known standard graph counterparts and we recover such matrices when $\mathcal{F}(v) \cong \mathbb{R}$ and $\mathcal{F}_{v \trianglelefteq e} v = 1$ for all $v \in V$ and $e \in E$.

**Vector bundles**. When restriction maps are orthogonal, we call the sheaf a *vector bundle*, a structure analogous to connections on Riemannian manifolds. For this reason, the sheaf Laplacian also takes the name *connection Laplacian* (Singer & Wu, 2012). The product $\mathcal{F}_{u \trianglelefteq e}^T \mathcal{F}_{v \trianglelefteq e}$ is then also orthogonal and is denoted $\mathbf{O}_{uv}$ referring to the transformation a vector undergoes when moved across a manifold via parallel transport. In this case we denote the node-stalk at $v$ by $\mathcal{B}(v)$, the bundle-adjacency by $\mathbf{A}_\mathcal{B}$ and the bundle Laplacian $\mathbf{L}_\mathcal{B}$, and its normalized version $\mathcal{L}_\mathcal{B} := \mathbf{I}_{dn \times dn} - \mathbf{D}_\mathcal{B}^{-1} \mathbf{A}_\mathcal{B}$.

Consider a $d$-dimensional vector field over the graph, i.e. a $d$-dimensional feature vector at each node denoted $\mathbf{X} \in \mathbb{R}^{nd}$ in which the signals are stacked column-wise. Similarly to the graph case, the operation $\mathbf{D}_\mathcal{B}^{-1} \mathbf{A}_\mathcal{B} \mathbf{X}$ is an averaging over the vector field, and $\mathcal{L}_\mathcal{B}$ a measure of smoothness, since:

$$\left(\mathbf{D}_\mathcal{B}^{-1} \mathbf{A}_\mathcal{B} \mathbf{X}\right)_u = \frac{1}{d_u} \sum_{u:(v,u) \in E} \mathbf{O}_{uv} \mathbf{x}_v \in \mathbb{R}^d, \text{ and } (\mathcal{L}_\mathcal{B} \mathbf{X})_u = \frac{1}{d_u} \sum_{u:(v,u) \in E} (\mathbf{x}_u - \mathbf{O}_{uv} \mathbf{x}_v) \in \mathbb{R}^d.$$

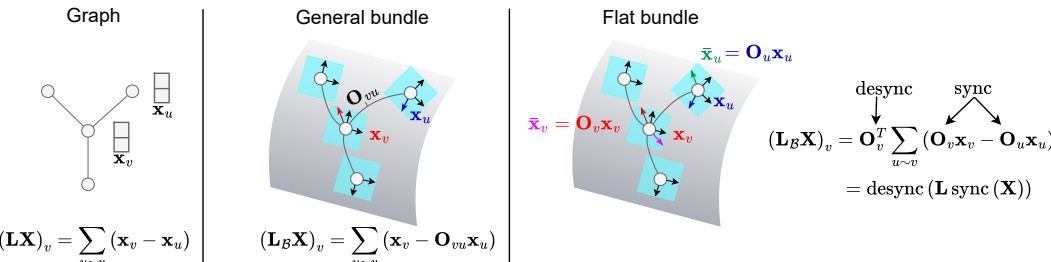

Figure 2: Comparison of different Laplacian and their actions on signals.

## 3 BUNDLE NEURAL NETWORKS

In this section, we derive BuNN from heat diffusion equations over flat vector bundles. We then discuss its relationship to GCN (Kipf & Welling, 2017) and NSD (Bodnar et al., 2022). We provide algorithmic implementations and additional practical details in the Appendix (Section E).

**Heat diffusion over bundles**. The *bundle Dirichlet energy* $\mathcal{E}_\mathcal{B}(\mathbf{X})$ of a vector field $\mathbf{X} \in \mathbb{R}^{nd}$ is:

$$\mathcal{E}_\mathcal{B}(\mathbf{X}) := \mathbf{X}^T \mathcal{L}_\mathcal{B} \mathbf{X} = \frac{1}{2} \sum_{(v,u) \in \mathsf{E}} \frac{1}{d_u} \|\mathbf{x}_u - \mathbf{O}_{uv}\mathbf{x}_v\|_2^2 .$$

The gradient of the Dirichlet energy $\nabla_\mathbf{X} \mathcal{E}_\mathcal{B}(\mathbf{X})$ is the random-walk Laplacian $\mathcal{L}_\mathcal{B}$. We can write down a *heat diffusion equation* over a vector bundle as a gradient flow, whose evolution equation with initial condition $\mathbf{X}(0) = \mathbf{X}$ satisfies $\partial_t \mathbf{X}(t) = -\mathcal{L}_\mathcal{B}\mathbf{X}(t)$. The solution to this equation can be written using matrix exponentiation as $\mathbf{X}(t) = \exp(-t\mathcal{L}_\mathcal{B})\mathbf{X}(0)$ (e.g. Hansen & Gebhart (2020)). We call the operator $\mathcal{H}_\mathcal{B}(t) := \exp(-t\mathcal{L}_\mathcal{B}) \in \mathbb{R}^{nd \times nd}$ the *bundle heat kernel*, which is defined as:

$$\mathcal{H}_\mathcal{B}(t) = \lim_{K \to \infty} \sum_{k=0}^{K} \frac{(-t\mathcal{L}_\mathcal{B})^k}{k!} .$$

Computing the heat kernel is necessary to solve the heat equation. An exact solution can be computed using spectral methods. For small $t$, one can instead consider the truncated Taylor expansion centered at 0, which amounts to fixing a $K$ in the above equation. Bodnar et al. (2022) instead approximates the solution using the Euler discretization of the heat equation with unit time step.

However, all these methods pose a challenge when the bundle structure is learned as in Bodnar et al. (2022), since the heat kernel has to be recomputed after every gradient update of the bundle structure. This high computational overhead limits the usability of Sheaf Neural Networks in applications.

**Flat vector bundles**. To address the scalability issues of general sheaves and vector bundles, we consider the special case of *flat vector bundles* in which every node $u$ gets assigned an orthogonal map $\mathbf{O}_u$, and every connection factorizes as $\mathbf{O}_{vu} = \mathbf{O}_v^T \mathbf{O}_u$. Consequently, the bundle Laplacian factors:

$$\mathcal{L}_\mathcal{B} = \mathbf{O}^T \left( \mathcal{L} \otimes \mathbf{I}_d \right) \mathbf{O},$$

where $\mathbf{O} \in \mathbb{R}^{nd \times nd}$ is block diagonal with $v$-th block being $\mathbf{O}_v$. We call the matrices $\mathbf{O}$ and $\mathbf{O}^T$ synchronization and desynchronization, respectively. We compare different Laplacians in Figure 2. This factorization avoids the $\mathcal{O}\left(d^3|\mathsf{E}|\right)$ cost of computing the restriction map over each edge. Additionally, Lemma 3.1 shows that it allows to cast the bundle heat equation into a standard graph heat equation. This reduces the computation of the bundle heat kernel to that of the cheaper graph heat kernel, an operator that does not change depending on the bundle and can, therefore, also be pre-computed.

**Lemma 3.1.** *For every node $v$, the solution at time $t$ of the heat equation on a connected bundle $\mathsf{G} = (\mathsf{V}, \mathsf{E}, \mathbf{O})$ with input node features $\mathbf{X}$ satisfies:*

$$(\mathcal{H}_\mathcal{B}(t)\mathbf{X})_v = \sum_{u \in \mathsf{V}} \mathcal{H}(t, v, u)\mathbf{O}_v^T \mathbf{O}_u \mathbf{x}_u,$$

*where $\mathcal{H}(t)$ is the standard graph heat kernel, and $\mathcal{H}(t, v, u) \in \mathbb{R}$ its the entry at $(v, u)$.*

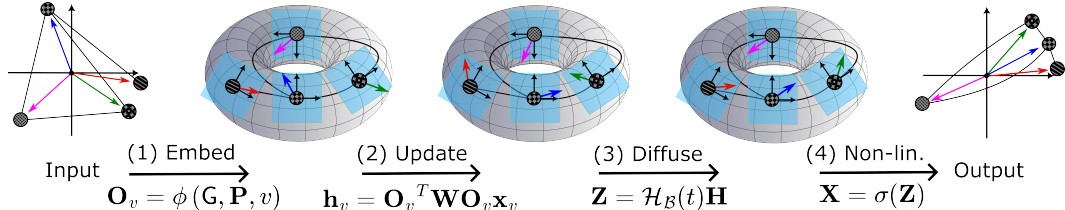

Figure 3: Example of the message diffusion framework on a graph with 4 nodes and 4 edges. From left to right: The input is a simple graph embedding with each color representing the feature vector at that node. (1) An orthogonal map is computed for each node in the graph by embedding the nodes in a continuous manifold with local reference frames (represented as a torus for visual aid), the features represented as colored vectors do not change. (2) The features are updated using learnable parameters $\mathbf{W}$. (3) The features are diffused for some time $t$ according to the heat equation on the manifold: a larger value of $t$ leads to a higher synchronization between all nodes as illustrated by the alignment of node features with respect to their local coordinates. (4) The output embedding is obtained by discarding the local coordinates and applying a non-linearity.

**The model**. The BuNN layer occurs in four steps, as illustrated in Figure 3. First, the bundle maps $\mathbf{O}_v$ are computed using a neural network $\phi$, the graph $\mathsf{G}$, positional encodings $\mathbf{P} \in \mathbb{R}^{n \times f}$ and the use of Householder reflections (Householder, 1958) or direct parameterization of the orthogonal group when $d = 2$. Second, an encoder step updates the node signals via a learnable matrix $\mathbf{W} \in \mathbb{R}^{d \times d}$, and bias $\mathbf{b} \in \mathbb{R}^d$. Next, the features are diffused over the *learned* vector bundle using the heat kernel. Finally, a non-linearity $\sigma$ is applied. We summarize the steps in the following equations:

$$\mathbf{O}_v^{(\ell)} := \phi^{(\ell)}(\mathsf{G}, \mathbf{P}, \mathbf{X}^{(\ell)}, v) \quad \forall v \in \mathsf{V} \tag{1}$$

$$\mathbf{h}_v^{(\ell)} := \mathbf{O}_v^{(\ell)T} \mathbf{W}^{(\ell)} \mathbf{O}_v^{(\ell)} \mathbf{x}_v^{(\ell)} + \mathbf{b}^{(\ell)} \quad \forall v \in \mathsf{V} \tag{2}$$

$$\mathbf{Z}^{(\ell+1)} := \mathcal{H}_{\mathcal{B}}(t) \mathbf{H}^{(\ell)} \tag{3}$$

$$\mathbf{X}^{(\ell+1)} := \sigma\left(\mathbf{Z}^{(\ell+1)}\right) \tag{4}$$

The diffusion time $t$ in Equation 3 is a hyperparameter determining the scale at which messages are diffused. For the case of small $t$, we approximate the heat kernel via its truncated Taylor series of degree $K$, and for large $t$, we use spectral methods. For simplicity of exposition, the steps above describe an update given a single bundle (i.e., $c = d$), meaning that $\mathbf{x}_v \in \mathbb{R}^d$. In general, we allow multiple bundles and vector field channels (Appendix E). Note that Equations 1, 2, and 3 are linear (or affine), and the non-linearities lie in 4. Equation 2 may be interpreted as a bundle-aware encoder, while Equation 3 is the *message diffusion* step guided by the heat kernel.

Without the bundle structure, Equation 3 would converge exponentially fast to constant node representations over the graph (e.g. Theorem 1 in Li et al. (2018)), potentially leading to over-smoothing. This is a limitation of existing diffusion-based GNNs (Xu et al., 2019a; Zhao et al., 2021). Accordingly, the bundle is crucial in this formulation to prevent node features from collapsing.

**Link to Graph Convolutional Networks**. It is possible to derive Graph Convolutional Networks (GCNs) (Kipf & Welling, 2017) as an approximation of BuNNs operating over a trivial bundle. Setting $t = 1$ Equation 3 becomes $\mathbf{Z}^{(l+1)} = \exp(-\mathcal{L}_{\mathsf{G}})\mathbf{H}^{(l)}$. The approximation $\exp(-\mathcal{L}_{\mathsf{G}}) \approx \mathbf{I} - \mathcal{L}_{\mathsf{G}}$ gives the update $\mathbf{Z}^{(l+1)} = (1 - \mathcal{L}_{\mathsf{G}})\mathbf{H}^{(l)} = \mathbf{A}_{\mathsf{G}}\mathbf{H}^{(l)}$ recovering the GCN update.

**Comparison with Sheaf Neural Networks**. Flat vector bundles are a special case of *cellular sheaves* (Curry, 2014; Bodnar et al., 2022), meaning that our model has close connections to Sheaf Neural Networks (SNNs) (Hansen & Gebhart, 2020; Bodnar et al., 2022; Barbero et al., 2022b;a). While most SNNs operate on fixed sheaves (Hansen & Gebhart, 2020; Barbero et al., 2022a; Battiloro et al., 2023), we focus on learning sheaves as in Neural Sheaf Diffusion (NSD) from Bodnar et al. (2022). BuNNs distinguish themselves from NSD in several ways. First, NSD approximates the heat equation using a time-discretized solution to the heat equation, which results in a standard message passing algorithm. In contrast, the direct use of the heat kernel allows BuNNs to *break away from the explicit message-passing paradigm*. This is in line with Battiloro et al. (2024) who define convolutions on manifolds using the heat kernel of fixed sheaves. Secondly, the use of flat bundles increases scalability since the bundle maps are computed at the node level. Additionally, flat bundles guarantees *path independence*, a requirement for the theory on the long time limit of NSD

in Bodnar et al. (2022) to hold, often not satisfied for general sheaf constructions such as ones used in NSD. Thirdly, we allow $\phi$ to be any GNN while NSD restricts $\phi$ to be an MLP. We found that incorporating the graph structure in $\phi$ improved the experimental results. The update in Equation 2 is also different to NSD in how the $\mathbf{W}$ and $\mathbf{b}$ are applied, and is necessary to prove our main theoretical result, Theorem 5.3. Additionally, Bodnar et al. (2022) experiments with general restriction maps as opposed to restricting them to orthogonal maps, and find that in practice orthogonal restrictions perform better, hence we consider orthogonal restriction maps. We provide experimental comparisons to NSDs in the experimental section and show that BuNNs significantly outperform their sheaf counterparts. We provide a summarized comparison between GCNs, SNNs, and BuNNs in Table 1.

**Comparison with other methods**. Another paradigm for learning on graphs is using graph transformers (GT), where every nodes communicate to each other through the use of self-attention. When the graph transformer is fully connected, all nodes communicate and therefore GT should not suffer from under-reaching or over-squashing, but might suffer from over-smoothing (Dovonon et al., 2024). When not fully connected, they might suffer from over-squashing (Barbero et al., 2024). Other paradigms such as Implicit Graph Neural Networks Fu et al. (2023) are also designed to tackle under-reaching and long-range dependencies while empirically not suffering from over-smoothing.

## 4    PROPERTIES OF BUNDLE NEURAL NETWORKS

We now give a formal analysis of the BuNN model. In Section 4.1, we derive the fixed points of the bundle heat diffusion, which the subspace of signals towards which solutions converges, and show that even in the limiting case BuNNs can retain information at the node level and therefore avoid over-smoothing. Section 4.2 discusses how our model can capture long-range interactions and mitigate over-squashing.

Table 1: Comparison between models in terms of message type and capability of mitigating issues related to GNNs.

|  | **GCN** | **SNN** | **BuNN** |
|---|---|---|---|
| **Propagation** | $\mathbf{A_G X}$ | $\mathcal{L_F X}$ | $\mathcal{H_B}(t)\mathbf{X}$ |
| **Message type** | Standard (Local) | Standard (Local) | Diffusive (Global) |
| **No under-reaching** | ✗ | ✗ | ✓ |
| **Alleviates over-squashing** | ✗ | ✗ | ✓ |
| **Alleviates over-smoothing** | ✗ | ✓ | ✓ |

### 4.1    FIXED POINTS AND OVER-SMOOTHING.

A major limitation of MPNNs is *over-smoothing*, where node features become indistinguishable as MPNN depth increases. This phenomenon is a major challenge for training deep GNNs. It arises because the diffusion in MPNNs resembles heat diffusion on graphs (Di Giovanni et al.), which converges to uninformative fixed points[1], leading to a loss of information at the node level. Bodnar et al. (2022) show that the richer bundle structure, however, gives rise to richer limiting behavior. Indeed, by Lemma 3.1, since $\lim_{t\to\infty} \mathcal{H}(t, v, u) = \frac{d_u}{2|E|}$, the limit over time of a solution is $\frac{1}{2|E|} \sum_{u\in V} d_u \mathbf{O}_v^T \mathbf{O}_u \mathbf{x}_u$. To understand the space of fixed points, notice that any $\mathbf{Y} \in \mathbb{R}^{n\times d}$ expressible as $\mathbf{y}_v = \frac{1}{2|E|} \sum_{u\in V} d_u \mathbf{O}_v^T \mathbf{O}_u \mathbf{x}_u$ for some $\mathbf{X}$ is a fixed point, and for any two nodes $u,\ v$:

$$\mathbf{O}_v \mathbf{y}_v = \frac{1}{2|E|} \sum_{w\in V} d_w \mathbf{O}_w \mathbf{x}_w = \mathbf{O}_u \mathbf{y}_u. \tag{5}$$

Consequently, the fixed points of vector diffusion have a global geometric dependency where all nodes relate to each other by some orthogonal transformation, e.g. the output in Figure 3. Equation 5 provides insight into the smoothing properties of BuNNs. When the bundle is trivial, the equation reduces to $\mathbf{y}_v = \mathbf{y}_u$, with no information at the node level, i.e. over-smoothing. When it is non-trivial, the output signal can vary across the graph (e.g., $\mathbf{y}_v \neq \mathbf{y}_u$ for two nodes $u$ and $v$). We summarize this in Proposition 4.1 and prove an implication in terms of over-smoothing in Proposition 4.2. Similar results showing that the signal can survive in deep layers as opposed to resulting in the constant signal have previously been done empirically and theoretically for other architectures, e.g. Chamberlain et al. (2021); Fu et al. (2023).

---

[1]Up to degree scaling if using the symmetric-normalized Laplacian, see Li et al. (2018).

**Proposition 4.1.** *Let* $\mathbf{Y}$ *be the output of a BuNN layer with* $t = \infty$, *where* $G$ *is a connected graph, and the bundle maps are not all equal. Then, there exists* $u, v \in \mathsf{V}$ *connected such that* $\mathbf{y}_v \neq \mathbf{y}_u$.

**Proposition 4.2.** *Let* $\mathbf{X}$ *be* 2-*dimensional features on a connected* $G$, *and assume that there are two nodes* $u$, $v$ *with* $\mathbf{p}_u \neq \mathbf{p}_v$. *Then there is an* $\delta > 0$ *such that for every* $K$, *there exist a* $K$-*layer deep BuNN with* 2 *dimensional stalks,* $t = \infty$, *ReLU activation,* $\phi^{(\ell)}$ *is a node-level MLP, and* $\|\mathbf{W}^{(\ell)}\| \leq 1$, *with output* $\mathbf{Y}$ *such that* $\|\mathbf{y}_v - \mathbf{y}_u\| > \delta$.

## 4.2 OVER-SQUASHING AND LONG RANGE INTERACTIONS.

While message-passing in MPNNs constitutes a strong inductive bias, it is problematic when the task requires the MPNN to capture interactions between distant nodes. These issues have been attributed mainly to the *over-squashing* problem. Topping et al. (2022) and Di Giovanni et al. (2023) formalize over-squashing through a sensitivity analysis, giving upper bounds on the sensitivity of the output at a node with respect to the input at an other. In particular, they show that under weak assumptions on the message function, the Jacobian of an MPNN satisfies the following inequality

$$|\partial \left(\text{MPNN}_\Theta(\mathbf{X})\right)_u / \partial \mathbf{x}_v| \leq c^\ell \left(\mathbf{A}^\ell\right)_{uv}, \tag{6}$$

for any nodes $u$, $v$, where $\ell$ is the depth of the network and $c$ is a constant. Given two nodes $u, v$ at a distance $r$, message-passing will require at least $r$ layers for the two nodes to communicate and overcome *under-reaching*. If $\ell$ is large, distant nodes can communicate, but Di Giovanni et al. (2023) show that over-squashing becomes dominated by vanishing gradients. Building on top of such sensitivity analysis, we compute the Jacobian for a BuNN layer in Lemma 4.3.

**Lemma 4.3.** *Let BuNN be a linear layer defined by Equations 1, 2 & 3 with hyperparameter* $t$. *Then, for any connected graph and nodes* $u$, $v$, *we have*

$$\frac{\partial \left(\text{BuNN}\left(\mathbf{X}\right)\right)_u}{\partial \mathbf{x}_v} = \mathcal{H}(t, u, v)\mathbf{O}_u^T \mathbf{W} \mathbf{O}_v.$$

The form of the Jacobian in Lemma 4.3 differs significantly from the usual form in Equation 6. First, all nodes communicate in a single BuNN layer since $\mathcal{H}(t, u, v) > 0$ for all $u, v$, and $t$, allowing for *direct pair-wise communication between nodes*, making a BuNN layer operate globally similarly to Transformer model, and therefore overcome under-reaching. Secondly, taking $t$ to be large allows BuNNs to operate on a larger scale, allowing stronger communication between distant nodes and overcoming over-squashing without the vanishing gradient problem.

Further, Lemma 4.3 gives a finer picture of the capabilities of BuNNs. For example, to mitigate over-squashing a node may decide to ignore information received from certain nodes while keeping information received from others. This allows the model to reduce the receptive field of certain nodes:

**Corollary 4.4.** *Consider* $n$ *nodes* $u$, $v$, *and* $w_i$, *for* $i = 1, \ldots n - 2$, *of a connected graph with* 2 *dimensional bundle such that* $\mathbf{p}_v \neq \mathbf{p}_{w_i}$ *and* $\mathbf{p}_u \neq \mathbf{p}_{w_i}$ $\forall i$. *Then in a BuNN layer with MLP* $\phi$, *at a given channel, the node* $v$ *can learn to ignore the information from all* $w_i s$ *while keeping information from* $u$.

## 5 EXPRESSIVITY OF THE MODEL

We now characterize the expressive power of BuNNs from a feature transformation perspective. This analysis extends that of Bodnar et al. (2022), who show that heat diffusion on sheaves can be expressive enough to linearly separate nodes in the infinite time limit. Instead, our results concern the more challenging problem of parameterizing arbitrary feature transformations and hold for finite time.

Most work on GNN expressivity characterize the ability of GNNs to distinguish isomorphism classes of graphs or nodes (Xu et al., 2019b; Morris et al., 2019; Azizian & Lelarge, 2021; Geerts & Reutter, 2022) or equivalently to approximate functions on them (Chen et al., 2019). These results typically rely on two major assumptions: 1) node features are fixed for each graph or are ignored completely; and 2) apply to a single graph or finitely many graphs of bounded size. Instead, our setting 1) includes features ranging in an infinite uncountable domain, and 2) includes infinite families of graphs. To this end, we define the notion of *compact uniform approximation* as a modification to that of uniform approximation from Rosenbluth et al. (2023) and we discuss our choice in Appendix C.

**Definition 5.1.** Let $\mathcal{F} \subseteq \mathcal{C}(\mathcal{G}, \mathbb{R}^c, \mathbb{R}^{c'})$ be a set of feature transformations over a family of graphs $\mathcal{G}$, and let $H \in \mathcal{C}(\mathcal{G}, \mathbb{R}^c, \mathbb{R}^{c'})$ a feature transformation over $\mathcal{G}$. We say that $\mathcal{F}$ *compactly uniformly approximates* $H$, if for all finite subsets $\mathcal{K} \subseteq \mathcal{G}$, for all compact $K \subset \mathbb{R}^c$, and for all $\epsilon > 0$, there exists an $F \in \mathcal{F}$ such that for all $\mathsf{G} \in \mathcal{K}$ and $\mathbf{X} \in K^{n_\mathsf{G}}$, we have that $||F_\mathsf{G}(\mathbf{X}) - H_\mathsf{G}(\mathbf{X})||_\infty \leq \epsilon$.

**Injective Positional Encodings (PEs).** In the case of a finite collection of graphs $\mathcal{G}$ and a fixed finite feature space $K$, the above definition reduces to the setting of Theorem 2 in Morris et al. (2019), since the node features are fixed to the finitely many values in $K$. However, when $K$ is not finite, the arguments in Morris et al. (2019) no longer work, as detailed in Appendix D. Consequently, Definition 5.1 subsumes graph-isomorphism testing, and it is, therefore, too strong for a polynomial-time GNN to satisfy. Hence, we will assume that the GNN has access to injective positional encodings, which we formally define in Definition D.1. This allows us to bypass the graph-isomorphism problem, while not trivializing the problem as we show next.

**Negative results with injective PE.** Characterizing the expressive power of GNNs in the uniform setting is an active area of research, with mostly negative results. Rosenbluth et al. (2024) prove negative results in the non-compact setting for both MPNNs with virtual-node and graph transformers, even with injective positional encodings. While extending their result to the compact setting is outside the scope of our work, we prove in Proposition 5.2 a negative result for bounded-depth MPNNs with injective PEs. Indeed, fixing the depth of the MPNNs to $\ell$, we can take a single compactly featured graph $\mathsf{G} \in \mathcal{G}$ with a diameter larger than $\ell$. As there is a node whose receptive field does not include all nodes in such a $\mathsf{G}$, the architecture cannot uniformly approximate every function on $\mathcal{G}$.[2]

**Proposition 5.2.** *There exists a family $\mathcal{G}$ consisting of connected graphs such that bounded-depth MPNNs are not compact uniform approximators, even if enriched with unique positional encoding.*

**Universality of BuNN.** In contrast to the negative results above, we now show that when equipped with injective PEs, BuNNs are universal with respect to compact uniform approximation. To the best of our knowledge, Theorem 5.3 is the first such positive feature approximation result for a GNN architecture, which demonstrates the remarkable modelling capabilities of BuNNs.

**Theorem 5.3.** *Let $\mathcal{G}$ be a possibly infinite set of connected graphs equipped with injective positional encodings. Then 2-layer deep BuNNs with encoder/decoder at each layer and $\phi^{(1)}$, $\phi^{(2)}$ being 2-layer deep MLP have compact uniform approximation over $\mathcal{G}$.*

In particular, let $\epsilon > 0$ and $H$ be a feature transformation on a finite subset $\mathcal{K} \subseteq \mathcal{G}$ and $K \subseteq \mathbb{R}^d$ a compact set, then there is a 2-layer deep BuNN with width of order $\mathcal{O}\left(\sum_{\mathsf{G} \in \mathcal{K}} |\mathsf{V}_\mathsf{G}|\right)$ that approximates $H$ over $\bigsqcup_{\mathsf{G} \in \mathcal{K}} K^{n_\mathsf{G}} \subseteq \bigsqcup_{\mathsf{G} \in \mathcal{G}} \mathbb{R}^{n_\mathsf{G} d}$. In other words, the required hidden dimension of BuNN is only linearly dependent on the number of nodes in the family of graphs.

## 6 EXPERIMENTS

In this section, we evaluate BuNNs through a range of synthetic and real-world experiments. We first validate the theory from Section 4 with two synthetic tasks. We then evaluate BuNNs on popular real-world benchmarks. We use truncated Taylor approximations for small values of $t$, while for larger $t$, we use the truncated spectral solution. We provide supplementary information on the implementation and precise experimental details in the Appendix (Sections E and F respectively).[3]

### 6.1 SYNTHETIC EXPERIMENTS: OVER-SQUASHING AND OVER-SMOOTHING

**Tasks**. In this experiment, we propose two new node-regression tasks in which nodes must average the input features of a subset of nodes. The input graph contains two types of nodes, whose features are sampled from disjoint distributions. The target for nodes is to output the average input feature over nodes of the other type and vice-versa, as illustrated in Figure 4. First, we test the capacity to mitigate **over-squashing**. We consider the case where the underlying graph is a barbell graph consisting of two fully connected graphs - each being a type and bridged by a single edge. This bottleneck makes it hard to transfer information from one cluster to another. Second, we test the capacity to mitigate **over-smoothing**. We consider the fully connected graph, in which all nodes are connected. The fully connected graph is a worst-case scenario for over-smoothing since after one step of message passing, the features are fully averaged over the graph and hence over-smoothed.

---

[2]This phenomenon in MPNNs is often called 'under-reaching'.

[3]All code can be found at https://github.com/jacobbamberger/BuNN

**Setup**. As a first baseline, we consider a constant predictor always predicting 0, the expected mean over the whole graph. As a second baseline, the cluster-specific constant predictor predicting the expected mean over the opposite cluster, that is, $\frac{\pm\sqrt{3}}{2}$ depending on the cluster.

Additionally, we consider GNN baselines to be a node-level MLP, GCN (Kipf & Welling, 2017), GraphSAGE (Hamilton et al., 2017), GAT (Veličković et al., 2018), NSD (Bodnar et al., 2022), and a fully connected GraphGPS (Rampášek et al. (2022)). The depth of MPNNs is fixed to the minimal depth to avoid under-reaching, namely 3 for the barbell and 1 for the fully connected graph, and ensure the width is large ($> 128$) considering the task. The depth of the transformer is tuned between 1 and 3 with 2 heads. We compare these to a BuNN with an MLP learning the bundle maps of a

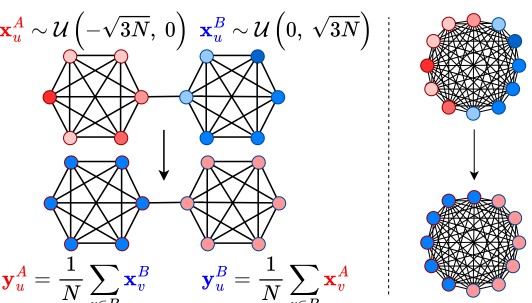

Figure 4: Synthetic over-squashing (left) and over-smoothing (right). In both cases, blue nodes output the average over the red nodes and vice-versa.

comparable number of parameters and the same depth. We use Adam optimizer with $10^{-3}$ learning rate, batch size 1, and train for 500 epochs. We use 100 samples for both training and testing.

**Results**. The results for $N = 10$ are reported in Table 2. All MPNNs perform poorly on the **over-squashing** task. All MPNNs perform comparably to baseline 2, showing their incapability to transfer information between clusters. This is explained by the fact that nodes from different clusters have high commute time and effective resistance, which tightly connects to over-squashing Di Giovanni et al. (2023); Black et al. (2023); Dong et al. (2024). On the other hand, BuNN achieves almost perfect accuracy on this task, which supports the claim that BuNN mitigates over-squashing. We note, however, that to solve the task perfectly, we need $t \geq 10$, allowing BuNN to operate on a larger scale more adapted to the task. To solve this task, BuNN can assign the orthogonal maps to separate the two types of nodes, making each node listen only to the nodes of the other type, a behavior proved to be possible in Corollary 4.4.

Similarly, the **over-smoothing** task on the clique graph is also challenging. Indeed, GCN and GAT perform exceptionally poorly, comparably to Baseline 1 which only has access to global information. Indeed, to the best of our knowledge, these are the only models with formal proofs of over-smoothing (Cai & Wang, 2020; Wu et al., 2023). GraphSAGE performs slightly better because it processes neighbors differently than it processes nodes themselves. Interestingly, GPS fails to generalize providing evidence for over-smoothing and over-squashing in graph transformers. Moreover, NSD and BuNN solve the task due to their capability to mitigate over-smoothing. Indeed, BuNN can learn to ignore nodes from a given cluster as proved in Corollary 4.4.

Table 2: **BuNN mitigates over-smoothing and over-squashing.** Mean squared error (MSE) of different models on the two synthetic tasks.

|  | Barbell *(over-squashing)* | Clique *(over-smoothing)* |
|---|---|---|
| Base. 1 | $30.97 \pm 0.42$ | $30.94 \pm 0.42$ |
| Base. 2 | $1.00 \pm 0.07$ | $0.99 \pm 0.08$ |
| MLP | $1.08 \pm 0.07$ | $1.10 \pm 0.08$ |
| GCN | $1.05 \pm 0.08$ | $29.65 \pm 0.34$ |
| SAGE | $0.90 \pm 0.29$ | $0.86 \pm 0.10$ |
| GAT | $1.07 \pm 0.09$ | $20.97 \pm 0.40$ |
| GPS | $1.06 \pm 0.13$ | $1.06 \pm 0.13$ |
| NSD | $1.09 \pm 0.15$ | $0.08 \pm 0.02$ |
| BuNN | $\mathbf{0.01 \pm 0.07}$ | $\mathbf{0.03 \pm 0.01}$ |

## 6.2 REAL-WORLD TASKS

We evaluate BuNNs on the Long Range Graph Benchmark (Dwivedi et al., 2022) and the heterophilic tasks from Platonov et al. (2023). We provide the implementation details in Appendix E.

**Heterophilic datasets**. As we have shown in Section 4, BuNNs are provably capable of avoiding over-smoothing. It is, therefore, natural to test how BuNN performs on heterophilic graphs where over-smoothing is recognized as an important limitation (e.g. Yan et al. (2022)). We follow their

Table 3: Results for the heterophilic tasks. Accuracy is reported for `roman-empire` and `amazon-ratings`, and ROC AUC is reported for `minesweeper`, `tolokers`, and `questions`. Best results are denoted by **bold**. Asterisk* denotes that some runs ran out of memory on an NVIDIA A10 GPU (24 GB).

|  | roman-empire | amazon-ratings | minesweeper | tolokers | questions |
|---|---|---|---|---|---|
| GCN | $73.69 \pm 0.74$ | $48.70 \pm 0.63$ | $89.75 \pm 0.52$ | $83.64 \pm 0.67$ | $76.09 \pm 1.27$ |
| SAGE | $85.74 \pm 0.67$ | $53.63 \pm 0.39$ | $93.51 \pm 0.57$ | $82.43 \pm 0.44$ | $76.44 \pm 0.62$ |
| GAT | $80.87 \pm 0.30$ | $49.09 \pm 0.63$ | $92.01 \pm 0.68$ | $83.70 \pm 0.47$ | $77.43 \pm 1.20$ |
| GAT-sep | $88.75 \pm 0.41$ | $52.70 \pm 0.62$ | $93.91 \pm 0.35$ | $83.78 \pm 0.43$ | $76.79 \pm 0.71$ |
| GT | $86.51 \pm 0.73$ | $51.17 \pm 0.66$ | $91.85 \pm 0.76$ | $83.23 \pm 0.64$ | $77.95 \pm 0.68$ |
| GT-sep | $87.32 \pm 0.39$ | $52.18 \pm 0.80$ | $92.29 \pm 0.47$ | $82.52 \pm 0.92$ | $78.05 \pm 0.93$ |
| NSD | $80.41 \pm 0.72$ | $42.76 \pm 0.54$ | $92.15 \pm 0.84$ | $78.83 \pm 0.76^*$ | $69.69 \pm 1.46^*$ |
| BuNN | $\mathbf{91.75 \pm 0.39}$ | $\mathbf{53.74 \pm 0.51}$ | $\mathbf{98.99 \pm 0.16}$ | $\mathbf{84.78 \pm 0.80}$ | $\mathbf{78.75 \pm 1.09}$ |

methodology to evaluate BuNN on the 5 heterophilic tasks proposed in Platonov et al. (2023). We run the models with 10 different seeds and report the mean and standard deviation of the test accuracy for `roman-empire` and `amazon-ratings`, and mean and standard deviation test ROC AUC for `minesweeper`, `tolokers`, and `questions`. We use the classical baselines from Platonov et al. (2023) and NSD, and provide the hyper-parameters in the Appendix (Section F.2).

**Results**. We report the results in Table 3. BuNN achieves the best score on all tasks, with an average relative improvement of $4.4\%$. Its score on `minesweeper` is particularly impressive, which is significantly ahead of the rest and for which BuNN solves the task perfectly. We found that the optimal value of $t$ over our grid search varies across datasets, being 1 for `amazon-ratings` and 100 for `roman-empire`. BuNN consistently outperforms the sheaf-based model NSD by a large margin, which we believe is due to the fact that NSD learns a map for every node-edge pairs making NSD more prone to overfitting, while BuNNs can act as a stronger regularizer. Such strong performance confirms our theory on over-smoothing from Section 4.1 and showcase the strong modeling capacity of BuNN in heterophilic settings.

Table 4: Results for the `Peptides-struct`, `Peptides-func`, and `PascalVOC-SP` tasks from the Long Range Graph Benchmark (results are $\times 100$ for clarity). The best result is **bold**.

| Model | Peptides-func Test AP ↑ | Peptides-struct Test MAE ↓ | PascalVOC-SP Test F1↑ |
|---|---|---|---|
| GCN | $68.60 \pm 0.50$ | $24.60 \pm 0.07$ | $20.78 \pm 0.31$ |
| GINE | $66.21 \pm 0.67$ | $24.73 \pm 0.17$ | $27.18 \pm 0.54$ |
| GatedGCN | $67.65 \pm 0.47$ | $24.77 \pm 0.09$ | $38.80 \pm 0.40$ |
| DReW | $71.50 \pm 0.44$ | $25.36 \pm 0.15$ | $33.14 \pm 0.24$ |
| SAN | $64.39 \pm 0.75$ | $25.45 \pm 0.12$ | $32.30 \pm 0.39$ |
| GPS | $65.34 \pm 0.91$ | $25.09 \pm 0.14$ | $\mathbf{44.40 \pm 0.65}$ |
| GAPH ViT | $69.42 \pm 0.75$ | $\mathbf{24.49 \pm 0.16}$ | - |
| Exphormer | $65.27 \pm 0.43$ | $24.81 \pm 0.07$ | $39.75 \pm 0.37$ |
| BuNN | $\mathbf{72.76 \pm 0.65}$ | $24.63 \pm 0.12$ | $40.49 \pm 0.46$ |

**Long Range Graph Benchmark**. In Section 4.2, we showed that BuNNs have desirable properties when it comes to over-squashing and modeling long-range interactions. To verify such claims empirically, we evaluate BuNN on tasks from the Long Range Graph Benchmark (LRGB) (Dwivedi et al., 2022). We consider the `Peptides` dataset consisting of $15\,535$ graphs which come with two associated graph-level tasks, `Peptides-func` and `Peptides-struct`, and the node classification on the `PascalVOC-SP` dataset with $11\,355$ graphs and $5.4$ million nodes where each graph corresponds to an image in Pascal VOC 2011 and each node to a superpixel in that image.

The graph classification task in `Peptides-func` is to predict the function of the peptide from 10 classes, while the regression task in `Peptides-struct` is inferring the 3D properties of the peptides. In both cases, we follow the standard experimental setup detailed by Dwivedi et al. (2022) alongside the updated suggestions from Tönshoff et al. (2023). The performance metric is Average

Precision (AP) for `Peptides-func` and Mean Absolute Error (MAE) for `Peptides-struct`. The node classification task in `PascalVOC-SP` consists of predicting a semantic segmentation label for each node out of 21 classes, with performance metric being the macro F1 score. We run each experiment on 4 distinct seeds and report mean and standard deviation over these runs. Baseline models are taken from Tönshoff et al. (2023) and include MPNNs, transformer models, and the current SOTA models (Gutteridge et al., 2023; He et al., 2023; Rampášek et al., 2022).

**Results**. We report the results in Table 4. BuNNs achieve, to the best of our knowledge, a new state-of-the-art result on `Peptides-func`. BuNNs also perform strongly on both other tasks, being second overall on `PascalVOC-SP` and clearly outperforming all MPNN models, and is third within one standard deviation of the second model on `Peptides-struct`. The overall strong performance on the LRGB benchmarks confirms our theory on oversquashing from Section 4.2 and provides further evidence of the long-range capabilities of BuNNs.

# 7 CONCLUSION

In this work, we proposed Bundle Neural Networks – a new type of GNN that operates via message diffusion on graphs. We gave a formal analysis of BuNNs showing that message diffusion can mitigate issues such as over-smoothing - since the heat equation over vector bundles admits a richer set of fixed points - and over-squashing - since BuNNs can operate at a larger scale than standard MPNNs. We also prove compact uniform approximation of BuNNs, a first expressivity result of its kind, characterizing their expressive power and establishing their superiority over MPNNs. We then confirmed our theory with carefully designed synthetic experiments. Finally, we showed that BuNNs perform well on heterophilic and long range tasks which are known to be challenging for MPNNs.

# 8 ACKNOWLEDGEMENTS

This research is partially supported by EPSRC Turing AI World-Leading Research Fellowship No. EP/X040062/1 and EPSRC AI Hub on Mathematical Foundations of Intelligence: An "Erlangen Programme" for AI No. EP/Y028872/1. X.D. acknowledges support from the Oxford-Man Institute of Quantitative Finance and the EPSRC (EP/T023333/1).

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

## A    Limitation and Future work

A limitation of our framework is that while message diffusion allows to operate on different scales of the graph, the computation of the heat kernel for large $t$ requires spectral methods and is therefore computationally expensive. An exciting research direction consists of using existing computational methods to approximate the heat kernel efficiently for large values of $t$. A limitation of our experiments is that we consider inductive graph regression/classification and transductive node regression/classification tasks but no link prediction task. A limitation in our theory is that Theorem 5.3 assumes injective positional encodings at the node level, which might only sometimes be available; future work could characterize the expressiveness when these are unavailable.

## B    Proofs

In this section, we provide proof of the theoretical results from the main text. Namely Lemma 3.1, Proposition 4.1, Lemma 4.3, Corollary 4.4, Proposition 5.2, and finally Theorem 5.3.

**Lemma 3.1.** *For every node $v$, the solution at time $t$ of heat diffusion on a connected bundle $\mathsf{G} = (\mathsf{V},\ \mathsf{E},\ \mathbf{O})$ with input node features $\mathbf{X}$ satisfies:*

$$\left(\mathcal{H}_{\mathcal{B}}(t)\mathbf{X}\right)_v = \sum_{u\in\mathsf{V}}\mathcal{H}(t,v,u)\mathbf{O}_v^T\mathbf{O}_u\mathbf{x}_u, \tag{7}$$

*where $\mathcal{H}(t)$ is the standard graph heat kernel, and $\mathcal{H}(t,\ v,\ u)\in\mathbb{R}$ its the entry at $(v,\ u)$.*

*Proof.* Since $\mathcal{L}_{\mathcal{B}} = \mathbf{O}_{\mathcal{B}}^T\mathcal{L}\mathbf{O}_{\mathcal{B}}$ we get $\mathcal{H}_{\mathcal{B}}(t,u,v) = \mathbf{O}_{\mathcal{B}}^T\mathcal{H}(t,u,v)\mathbf{O}_{\mathcal{B}}$ by the definition of the heat kernel. □

### B.1    Over-smoothing: proofs of Section 4.1.

In this section, we prove the results on the stable states and on over-smoothing. The first result follows straightforwardly from Lemma 3.1.

**Proposition 4.1.** *Let $\mathbf{Y}$ be the output of a BuNN layer with $t = \infty$, where $G$ is a connected graph, and the bundle maps are not all equal. Then, $u,v \in \mathsf{V}$ is connected such that $\mathbf{y}_v \neq \mathbf{y}_u$ almost always.*

*Proof.* Consider a stable signal $\mathbf{Y} \in \ker \mathcal{L}_{\mathcal{B}}$ and pick $u,v \in \mathsf{V}$ such that $\mathbf{O}_u \neq \mathbf{O}_v$. As $\mathbf{Y}$ is stable, it must have $0$ bundle Dirichlet energy, so we must have that $\mathbf{O}_u\mathbf{y}_u = \mathbf{O}_v\mathbf{y}_v$, but as $\mathbf{O}_u \neq \mathbf{O}_v$ we have that $\mathbf{y}_u \neq \mathbf{y}_v$, which holds except in degenerate cases such as when the matrices $\mathbf{O}_u$ are reducible, or when the original signal is the zero vector. □

The idea of the second result is that due to the injectivity of the positional encodings on the two nodes, we can set their restriction maps so that the first layer zeroes out the first channel of one node and the second channel of the second. The next $K - 1$ layers simply keep this signal fixed by realizing them as a fixed point of the bundle heat equation.

**Proposition 4.2.** *Let $\mathbf{X}$ be $2$-dimensional features on a connected $\mathsf{G}$, and assume that there are two nodes $u,\ v$ with $\mathbf{p}_u \neq \mathbf{p}_v$. Then there is an $\delta > 0$ such that for every $K$, there exist a $K$-layer deep BuNN with $2$ dimensional stalks, $\phi^{(\ell)}$ is a node-level MLP, ReLU activation, and $\|\mathbf{W}^{(\ell)}\| \leq 1$, with output $\mathbf{Y}$ such that $\|\mathbf{y}_v - \mathbf{y}_u\| > \delta$.*

*Proof.* We will show the result for the limit $t \to \infty$. Let $\mathbf{O}_w = \mathbf{Id}$ for $w \neq u$ and $\mathbf{O}_v = \begin{pmatrix} 0 & 1 \\ 1 & 0 \end{pmatrix}$. These restriction maps can be learned by assumption since $\mathbf{p}_u \neq \mathbf{p}_v$ and MLPs are universal. Define

$$\mathbf{h} = \sum_{w\neq u}\frac{d_w}{2|\mathsf{E}|}\mathbf{O}_w\mathbf{x}_w + \frac{d_u}{2|\mathsf{E}|}\begin{pmatrix} 0 & 1 \\ 1 & 0 \end{pmatrix}\mathbf{x}_u \in \mathbb{R}^2$$

and denote $h_0$ the entry in the first dimension and $h_1$ its second entry.

**Case 1:** $(h_0)^2 > 0$ **or** $(h_1)^2 > 0$**.** We start with the case $(h_0)^2 > 0$. Set $\delta = (h_0)^2$. If $h_0 > 0$, set the first weight matrix to be $\mathbf{W}^{(1)} = \begin{pmatrix} 1 & 0 \\ 0 & 0 \end{pmatrix}$, otherwise let $\mathbf{W}^{(1)} = \begin{pmatrix} -1 & 0 \\ 0 & 0 \end{pmatrix}$. Set the first bias to $\mathbf{0}$. The output before the activation will then be $\mathbf{h}^{(1)} = (|h_0|,\ 0)^T$ for all $w \neq u$ and $\mathbf{h}^{(1)} = (0,\ |h_0|)^T$ by Equation 5. Since $\sigma$ is ReLU, it does not change the output. For the next layers, picking the same restriction maps $\mathbf{O}_w$s and $\mathbf{W}^{(\ell)} = \mathbf{Id}$ also do not change the output. Hence at layer $K$, the output at $v$ is $\mathbf{y}_v = (|h_0|,\ 0)^T$ and at $u$ it is $\mathbf{y}_u = (0\ ,|h_0|)^T$. We conclude since $\|\mathbf{y}_u - \mathbf{y}_u\|_2^2 = 2|h_0|^2 \geq \delta$. If $(h_0)^2 = 0$ and $(h_1)^2 = 0$ the argument is the same.

**Case 2:** $(h_0)^2 = 0$ **and** $(h_1)^2 = 0$**.** Set $\delta = \frac{1}{2}$ and $\mathbf{W}^{(1)} = \mathbf{0}$ and $\mathbf{b}^{(1)} = (1,\ 0)^T$. Before the message diffusion step, the signal is constant on the nodes and equal to $\mathbf{b}$. By Equation 5, the pre-activation output of the first layer is $\mathbf{b} = (1,\ 0)$ at all nodes $w \neq u$ and $\mathbf{O}_u^T \mathbf{b} = (0,\ 1)^T$ at $u$. Since all entries are positive, the ReLU activation leaves them fixed. The subsequent layer can be set as in the previous case to keep the node embedding fixed. Consequently, $\|\mathbf{y}_u - \mathbf{y}_v\| = 1 > \frac{1}{2}$. $\qquad\square$

### B.2 Over-squashing: proofs of Section 4.2.

In this section we prove our results on over-squashing and long-range interactions. The Jacobian result follows straightforwardly from Lemma 3.1 and the definition of a BuNN layer, and the Corollary follows from the Jacobian result.

**Lemma 4.3.** *Let BuNN be a linear layer defined by Equations 1, 2 & 3 with hyperparameter $t$. Then, for any connected graph and nodes $u$, $v$, we have*

$$\frac{\partial \left(\mathrm{BuNN}\left(\mathbf{X}\right)\right)_u}{\partial \mathbf{x}_v} = \mathcal{H}(t,u,v)\mathbf{O}_u^T \mathbf{W} \mathbf{O}_v,$$

*and therefore*

$$\lim_{t \to \infty} \frac{\partial \left(\mathrm{BuNN}\left(\mathbf{X}\right)\right)_u}{\partial \mathbf{x}_v} = \frac{d_v}{2|\mathsf{E}|}\mathbf{O}_u^T \mathbf{W} \mathbf{O}_v.$$

*Proof.* The result follows from the closed-form solution of the heat kernel from Lemma 3.1. We start by applying the bundle encoder from Equation 2 that updates each node representation as $\mathbf{h}_v = \mathbf{O}_v^T \mathbf{W} \mathbf{O}_v \mathbf{x}_v + \mathbf{b}$. Since the $\mathbf{O}_u$ do not depend on the signal $\mathbf{X}$, we get

$$\frac{\partial \left(\mathrm{BuNN}\left(\mathbf{X}\right)\right)_u}{\partial \mathbf{x}_v} = \frac{\partial}{\partial \mathbf{x}_v}\left[\sum_{v \in \mathsf{V}} \mathcal{H}(t,u,v)\mathbf{O}_u^T \mathbf{O}_v \left(\mathbf{O}_v^T \mathbf{W} \mathbf{O}_v \mathbf{x}_v + \mathbf{b}\right)\right] \tag{8}$$

$$= \mathcal{H}(t,u,v)\mathbf{O}_u^T \mathbf{W} \mathbf{O}_v. \tag{9}$$

The second statement follows from the fact that $\mathcal{H}(t,u,v) \to \frac{d_u}{2|E|}$ $\qquad\square$

To illustrate the flexibility of such a result, we examine a setting in which we want nodes to select which nodes they receive information from, therefore 'reducing' their receptive field.

**Corollary 4.4.** *Consider $n$ nodes $u$, $v$, and $w_i$, for $i = 1, \ldots n - 2$, of a connected graph with 2 dimensional bundle such that $\mathbf{p}_v \neq \mathbf{p}_{w_i}$ and $\mathbf{p}_u \neq \mathbf{p}_{w_i}\ \forall i$. Then in a BuNN layer with MLP $\phi$, at a given channel, the node $v$ can learn to ignore the information from all $w_i$s while keeping information from $u$.*

*Proof.* We denote $\mathbf{y}$ the output of the layer, and index the two dimensions by super-scripts, i.e. $\mathbf{y} = \begin{pmatrix} \mathbf{y}^{(1)} \\ \mathbf{y}^{(2)} \end{pmatrix}$. Our goal is to have $\frac{\partial \mathbf{y}_v^{(1)}}{\partial \mathbf{y}_u} \neq (0 \quad 0)$, while $\frac{\partial \mathbf{y}_v^{(1)}}{\partial \mathbf{y}_{w_i}} = (0 \quad 0)$ for all $i$. This would make the first channel of the output at $v$ insensitive to the input at all $w_i$s while being sensitive to the input at node $u$.

Fix $\mathbf{O}_v = \mathbf{O}_u = \begin{pmatrix} 0 & 1 \\ 1 & 0 \end{pmatrix}$ and $\mathbf{O}_{w_i} = \begin{pmatrix} 1 & 0 \\ 0 & 1 \end{pmatrix}$. Such maps can always be learned by an MLP, by the assumptions on $\mathbf{p}_v$, $\mathbf{p}_u$, and $\mathbf{p}_{w_i}$ and by the universality of MLPs. Let the weight matrix be

$\mathbf{W} = \begin{pmatrix} w_{11} & w_{12} \\ w_{21} & w_{22} \end{pmatrix}$. By Lemma 4.3 we get $\frac{\partial \mathbf{y}_v}{\partial \mathbf{x}_u} = \mathcal{H}(t,v,u)\mathbf{O}_v^T \mathbf{W}\mathbf{O}_u = \mathcal{H}(t,v,u)\begin{pmatrix} w_{22} & w_{12} \\ w_{21} & w_{11} \end{pmatrix}$

and $\frac{\partial \mathbf{y}_v}{\partial \mathbf{x}_{w_i}} = \mathcal{H}(t,v,w_i)\mathbf{O}_v^T \mathbf{W}\mathbf{O}_{w_i} = \mathcal{H}(t,v,w_i)\begin{pmatrix} w_{21} & w_{22} \\ w_{11} & w_{12} \end{pmatrix}$. Setting $w_{21}$ and $w_{22}$ to $0$ gives

$\frac{\partial \mathbf{y}_v^{(1)}}{\partial \mathbf{x}_{w_i}} = (0 \quad 0)$ and $\frac{\partial \mathbf{y}_v^{(1)}}{\partial \mathbf{x}_u} = \mathcal{H}(t,v,u)(0 \quad w_{12}) \neq \mathbf{0}$, as desired. $\qquad \square$

### B.3 EXPRESSIVITY OF BuNNs: PROOFS OF SECTION 5.

We now turn to BuNN's expressivity. Before proving that BuNNs have compact uniform approximation, we prove that MPNNs fail to have this property. This proves BuNNs' superiority and shows that uniform expressivity is a good theoretical framework for comparing GNN architectures.

**Proposition 5.2.** *There exists a family $\mathcal{G}$ consisting of connected graphs such that bounded-depth MPNNs are not compact uniform approximators, even if enriched with unique positional encoding.*

*Proof.* Let $\mathcal{G}$ be any family of connected graphs with an unbounded diameter (for example, the $n \times n$ grids with $n \to \infty$). Let the depth of the MPNN be $L$. Let $\mathsf{G} \in \mathcal{G}$ be a graph with diameter $> L$, and let $u$ and $v$ be two nodes in $\mathsf{V}_\mathsf{G}$ at distance $> L$. Note that the output at $v$ will be insensitive to the input at $u$, and therefore, the MPNN cannot capture feature transformations where the output at $v$ depends on the input at $u$. This argument holds even when nodes are given unique positional encodings. $\qquad \square$

We now turn to our main theoretical contribution. The proof of Theorem 5.3 is split into two parts. The first proves that 1-layer BuNNs have compact uniform approximation over *linear feature transformations*. The second part is extending to continuous feature transformation, which is an application of classical results.

We start by recalling the definition of a linear feature transformation over a family of graphs $\mathcal{G}$:

**Definition B.1.** A linear feature transformation $L \in \mathcal{C}(\mathcal{G}, \mathbb{R}^c, \mathbb{R}^{c'})$ over a family of graphs $\mathcal{G}$ is an assignment of each graph $\mathsf{G} \in \mathcal{G}$ to a linear map $L_\mathsf{G} : \mathbb{R}^{n_\mathsf{G} c} \to \mathbb{R}^{n_\mathsf{G} c'}$. Here, linearity means that for any two node-signals $\mathbf{X}_1 \in \mathbb{R}^{nc}$ and $\mathbf{X}_2 \in \mathbb{R}^{nc'}$, and any real number $\alpha \in \mathbb{R}$, it holds that $L_\mathsf{G}(\alpha \mathbf{X}_1) = \alpha L_\mathsf{G}(\mathbf{X}_1)$, and $L_\mathsf{G}(\mathbf{X}_1 + \mathbf{X}_2) = L_\mathsf{G}(\mathbf{X}_1) + L_\mathsf{G}(\mathbf{X}_2)$.

We will need the following Theorem, which adapts classical results on the universality of MLPs.

**Theorem B.2.** *If a class of neural networks has compact uniform approximation over $\mathcal{G}$ with respect to linear functions and contains non-polynomial activations, then it has compact universal approximation over $\mathcal{G}$ with respect to continuous functions.*

*Proof.* Classical theorems such as Theorem 1 in (Cybenko, 1989) allow us to approximate any continuous function over a compact set in a finite dimensional vector space by composing a linear map $\mathbf{C}$, an activation $\sigma$, and an affine map $\mathbf{A} \cdot + \mathbf{b}$. Given a finite family of graph $\mathcal{G}$, the space of node features on all graphs is a finite dimensional vector space. By assumption, we can implement the linear map, the activation, and the affine map. Hence, by composing them, we can approximate any continuous function over the compact set. $\qquad \square$

We are now ready to prove the paper's main result: that, given injective positional encodings, BuNNs are compact universal approximators of feature transformations.

**Theorem 5.3.** *Let $\mathcal{G}$ be a set of connected graphs with injective positional encodings. Then 2-layer deep BuNNs with encoder/decoder at each layer and $\phi$ being a 2-layer MLP have compact uniform approximation over $\mathcal{G}$.*

*In particular, let $\epsilon > 0$ and $h$ be a feature transformation supported on $\bigsqcup_{\mathsf{G} \in \mathcal{K}} K^{n_\mathsf{G}} \subseteq \bigsqcup_{\mathsf{G} \in \mathcal{G}} \mathbb{R}^{n_\mathsf{G} d}$ with $\mathcal{K} \subseteq \mathcal{G}$ finite and $K \subseteq \mathbb{R}^d$ a compact set, then there is a 2-layer deep BuNN with width $\mathcal{O}\left(\sum_{\mathsf{G} \in \mathcal{K}} |\mathsf{V}_\mathsf{G}|\right)$ that approximates $h$ with uniform error $< \epsilon$.*

*Proof.* **Reducing to linear approximation.** It suffices to show that a BuNN layer can approximate any linear feature transformation $L$ because we can apply classical results such as Theorem B.2

to get universal approximation of 2-layer deep networks with activation. Following Definition 5.1, we aim to show that we can approximate a linear feature transformation $L$ on any compact subset. For this, we fix $\epsilon > 0$, the finite subset $\mathcal{K} \subseteq \mathcal{G}$, and compact feature space $K \subseteq \mathbb{R}^c$. In fact, we assume that $K = \mathbb{R}^c$ since approximating a linear map on any compact feature space is equivalent to approximating it on the whole space because a linear map defined on a neighborhood of the $0$ vector can be extended uniquely to the whole vector space. Our goal is therefore to find a parameterization of a single BuNN layer such that for any graph $\mathsf{G} \in \mathcal{K}$ and for any input feature $\mathbf{X} \in \mathbb{R}^{n_{\mathsf{G}} c}$, we have $\|L_{\mathsf{G}}(X) - \text{BuNN}_{\mathsf{G}}(\mathbf{X})\|_\infty < \epsilon$. We will show that $L$ can be parameterized exactly. Since $L$ is linear, it suffices to find a linear BuNN layer that satisfies for any $\mathsf{G} \in \mathcal{K}$ and any $\mathbf{X} \in \mathbb{R}^{n_{\mathsf{G}} c}$, $\frac{\partial (L(\mathbf{X}))_u}{\partial \mathbf{x}_v} = \frac{\partial (\text{BuNN} \mathbf{X})_u}{\partial \mathbf{x}_v}$. By Lemma 4.3, we have $\frac{\partial \text{BuNN}(\mathbf{X})_u}{\partial \mathbf{x}_v} = \mathcal{H}(t, u, v) \mathbf{O}_u \mathbf{W} \mathbf{O}_v^T$. Hence, since MLPs are universal and the positional encodings are injective, it suffices to find bundle maps $\mathbf{O} : \bigsqcup_{\mathsf{G} \in \mathcal{K}} V_{\mathsf{G}} \to O(k)$ and $\mathbf{W}$ such that $\frac{1}{n_{\mathsf{G}} d_u} \sum_{v \in \mathsf{V}} \mathbf{O}_u^T \mathbf{W} \mathbf{O}_v = \frac{\partial (LX)_u}{\partial X_v}$ for every $u, v \in \mathsf{G}$ and every $\mathsf{G} \in \mathcal{K}$.

**Defining the encoder and decoder:** In order to find such a BuNN, we first need a linear encoder $\text{lift} : \mathbb{R}^c \to \mathbb{R}^{2ck}$ which will be applied at every node before applying a $2ck$ dimensional BuNN layer. The lifting transformation maps each node vector $\mathbf{X}_u$ to the concatenation of $k$ vectors $\mathbf{X}_u$ interleaved with $k$ vectors $\mathbf{0} \in \mathbb{R}^c$. This is equivalent to the linear transformation given by left multiplication by $(\mathbf{I}_{c \times c}, \mathbf{0}, \ldots, \mathbf{I}_{c \times c}, \mathbf{0})^T \in \mathbb{R}^{2ck \times c}$. After the $2ck$ dimensional BuNN network, we will also need a linear decoder $\text{pool} : \mathbb{R}^{2ck} \to \mathbb{R}^c$ applied to every node individually, which is the sum of the $k$ different $c$-dimensional vectors that are at even indices. This is equivalent to left multiplication by the matrix $(\mathbf{I}_{c \times c}, \mathbf{0}_{c \times c}, \ldots, \mathbf{I}_{c \times c}, \mathbf{0}_{c \times c}) \in \mathbb{R}^{c \times 2ck}$. These two can be seen as a linear encoder and linear decoder, often used in practical GNN implementations. We prove the result by adding the lifting and pooling layers and using the higher dimensional $\widehat{BuNN}$ layer, i.e. we prove that $\text{BuNN} = \text{pool} \circ \widehat{\text{BuNN}} \circ \text{lift}$ can approximate any linear maps which satisfy the encoder and decoder assumption of the Theorem statement.

**Defining the 'universal bundle':** We fix $k = \sum_{\mathsf{G} \in \mathcal{K}} |V_{\mathsf{G}}|$, so we can interpret our embedding space as a lookup table where each index corresponds to a node $v \in \bigsqcup_{\mathsf{G} \in \mathcal{K}} V_{\mathsf{G}}$. In turn, we can think of the parameter matrix $\mathbf{W} \in \mathbb{R}^{(\sum_{G \in \mathcal{K}} |V_{\mathsf{G}}|) \times (\sum_{G \in \mathcal{K}} |V_{\mathsf{G}}|)}$ as a lookup table where each entry corresponds to a pair of nodes in our dataset $\mathcal{K}$. Still thinking of the indices of the $2ck$ dimensions as $2c$-dimensional vectors indexed by the $k$ nodes in our dataset, we define $\mathbf{O}_u \in O(2ck)$ as a block diagonal matrix with $k$ different $2c$-dimensional blocks, where the $k_i$th block is denoted $\mathbf{O}_u^{k_i}$. These are all set to the identity except for the block at the index corresponding to node $u$, which is defined as $\begin{pmatrix} \mathbf{0_{c \times c}} & \mathbf{I}_{c \times c} \\ \mathbf{I}_{c \times c} & \mathbf{0_{c \times c}} \end{pmatrix}$ which is a $2c \times 2c$ matrix that acts by permuting the first $c$ dimensions with the second $c$ dimensions.

**Computing the partial derivatives.** Since our model BuNN is a composition of linear maps, and since the maps $pool$ and $lift$ are applied node-wise, we get

$$\frac{\partial (\text{BuNN}(\mathbf{X}))_u}{\partial \mathbf{x}_v}$$
$$= \text{pool} \frac{\partial \left( \widehat{\text{BuNN}}(\text{lift}(\mathbf{X})) \right)_u}{\partial \text{lift}(\mathbf{X}_v)} \text{lift}$$
$$= (\mathbf{I}_{c \times c}, \mathbf{0}_{c \times c}, \ldots, \mathbf{I}_{c \times c}, \mathbf{0}_{c \times c}) \mathcal{H}(t, u, v) \mathbf{O}_u^T \mathbf{W} \mathbf{O}_v (\mathbf{I}_{c \times c}, \mathbf{0}_{c \times c}, \ldots, \mathbf{I}_{c \times c}, \mathbf{0}_{c \times c})^T$$
$$= \mathcal{H}(t, u, v) \sum_{1 \leq k_1, k_2 \leq k} (\mathbf{I}_{c \times c}, \mathbf{0}_{c \times c}) \mathbf{O}_u^{k_1 T} \mathbf{W}^{k_1 k_2} \mathbf{O}_v^{k_2} (\mathbf{I}_{c \times c}, \mathbf{0}_{c \times c})^T$$

We proceed by partitioning the indexing by $(k_1, k_2)$ into four cases. The first case is $C_1 = \{(k_1, k_2) \text{ such that } (k_1 \neq u, v \text{ and } k_2 \neq u, v)\}$ for which both $\mathbf{O}_u^{k_1}$ and $\mathbf{O}_u^{k_2}$ act like the identity. The second case is $C_2 = \{(k_1, k_2) \text{ such that } k_1 = u \text{ and } k_2 \neq v\}$ where $\mathbf{O}_u^{k_1}$ flips the first $c$ rows with the second $c$ rows and $\mathbf{O}_v^{k_2}$ acts like the identity. $C_3 = \{(k_1, k_2) \text{ such that } k_2 = v \text{ and } k_1 \neq u\}$ where $\mathbf{O}_v^{k_2}$ flips the first $c$ columns with the second $c$ columns, and $\mathbf{O}_u^{k_1}$ acts like the identity on the rows. Finally, the last case is when $k_1 = u$ and $k_2 = v$ in which $\mathbf{O}_u^{k_1}$ flips the rows, and $\mathbf{O}_v^{k_1}$ flips the columns. Note that by the injectivity assumption as defined in Definition D.1, each node has

a unique positional encoding and therefore there such matrices can be parameterized for all nodes simultaneously.

$$
\begin{aligned}
\ldots &= \mathcal{H}(t,u,v) \sum_{1 \leq k_1,\, k_2 \leq k} (\mathbf{I}_{c \times c}, \mathbf{0}_{c \times c}) \, \mathbf{O}_u^{k_1 T} \mathbf{W}^{k_1 k_2} \mathbf{O}_v^{k_2} (\mathbf{I}_{c \times c}, \mathbf{0}_{c \times c})^T \\
&= \mathcal{H}(t,u,v) \, (\mathbf{I}_{c \times c}, \mathbf{0}_{c \times c}) \left[ \sum_{(k_1,k_2) \in C_1} \begin{pmatrix} \mathbf{W}_{00}^{k_1 k_2} & \mathbf{W}_{01}^{k_1 k_2} \\ \mathbf{W}_{10}^{k_1 k_2} & \mathbf{W}_{11}^{k_1 k_2} \end{pmatrix} + \sum_{(k_1,k_2) \in C_2} \begin{pmatrix} \mathbf{W}_{10}^{k_1 k_2} & \mathbf{W}_{11}^{k_1 k_2} \\ \mathbf{W}_{00}^{k_1 k_2} & \mathbf{W}_{01}^{k_1 k_2} \end{pmatrix} \right. \\
&\qquad \left. + \sum_{(k_1,k_2) \in C_3} \begin{pmatrix} \mathbf{W}_{01}^{k_1 k_2} & \mathbf{W}_{00}^{k_1 k_2} \\ \mathbf{W}_{11}^{k_1 k_2} & \mathbf{W}_{10}^{k_1 k_2} \end{pmatrix} + \begin{pmatrix} \mathbf{W}_{11}^{uv} & \mathbf{W}_{10}^{uv} \\ \mathbf{W}_{01}^{uv} & \mathbf{W}_{00}^{uv} \end{pmatrix} \right] (\mathbf{I}_{c \times c}, \mathbf{0}_{c \times c})^T \\
&= \mathcal{H}(t,u,v) \left[ \sum_{(k_1,k_2) \in C_1} \mathbf{W}_{00}^{k_1 k_2} + \sum_{(k_1,k_2) \in C_2} \mathbf{W}_{10}^{k_1 k_2} + \sum_{(k_1,k_2) \in C_3} \mathbf{W}_{01}^{k_1 k_2} + \mathbf{W}_{11}^{uv} \right]
\end{aligned}
$$

Where the last line is obtained by applying $(\mathbf{I}_{c \times c}, \mathbf{0}_{c \times c})$ on the left and $(\mathbf{I}_{c \times c}, \mathbf{0}_{c \times c})^T$ on the right, an operation that selects the upper left $c \times c$ block. We observe that setting all $\mathbf{W}_{00}^{k_1 k_2} = \mathbf{W}_{01}^{k_1 k_2} = \mathbf{W}_{10}^{k_1 k_2}$ to $\mathbf{0}_{c \times c}$ and setting $\mathbf{W}_{11}^{uv} := \frac{1}{\mathcal{H}(t,u,v)} \frac{\partial (L\mathbf{X})_u}{\partial \mathbf{x}_v}$ if the nodes corresponding to $u$ and $v$ lie in the same graph and $\mathbf{0}_{c \times c}$ otherwise. This allows us to conclude that any linear layer can be parameterized, completing the proof of the theorem. $\qquad\square$

## C    DISCUSSION ON COMPACT UNIFORM APPROXIMATION VERSUS UNIFORM APPROXIMATION

A strong definition of expressivity that deals with infinite collections of graphs was proposed in Rosenbluth et al. (2023). This definition subsumes graph-isomorphism testing (where the input feature on graphs is constant). Furthermore, it also deals with infinite families of graphs, as opposed to most mainstream theorems of GNN expressivity, which are proved for graphs of bounded size (e.g. Azizian & Lelarge (2021); Geerts & Reutter (2022)). See Section 2 for the notation and definition of features transformations.

**Definition C.1** (From Rosenbluth et al. (2023)). Let $c$, $c' \in \mathbb{N}$ and take $\mathbb{R}$ as feature space. Consider a collection of graphs $\mathcal{G}$. Let $\Omega \subseteq \mathcal{C}\left(\mathcal{G}, \mathbb{R}^c, \mathbb{R}^{c'}\right)$ be a set of feature transformations over $\mathcal{G}$, and let $H \in \mathcal{C}\left(\mathcal{G}, \mathbb{R}^c, \mathbb{R}^{c'}\right)$ a feature transformation over $\mathcal{G}$. We say that $\Omega$ *uniformly additively approximates* $H$, notated $\Omega \approx H$ if $\forall \epsilon > 0 \ \forall$ compact $K^n \subset \mathbb{R}^{nc} \ \exists F \in \Omega$ such that:, $\forall \mathsf{G} \in \mathcal{G} \ \forall X \in K^{n_\mathsf{G} c} \ \|F_\mathsf{G}(X) - H_\mathsf{G}(X)\|_\infty \leq \epsilon$ where the sup norm $\|\cdot\|_\infty$ is taken over all nodes and dimensions of $n_\mathsf{G} c'$ dimensional output.

Note that this definition differs from our Definition 5.1 in that it requires uniform approximation over all graphs in $\mathcal{G}$ simultaneously, while we allow the width to vary with the finite subset $\mathcal{K} \subseteq \mathcal{G}$, similar to how classical results allow the width to vary with the compact set over which to approximate the function. Such a definition has proven useful in Rosenbluth et al. (2023) to distinguish different aggregation functions and in Rosenbluth et al. (2024) to distinguish MPNNs with virtual nodes from Transformers. However, we argue that **the definition above is too strong for a finite parameter GNN**. This is because it requires uniform approximation over a *non-compact set*, which contrasts with traditional work on expressivity and is generally unfeasible and impractical. Indeed, finite-parameters MLPs are not universal over the whole domain $\mathbb{R}$ under the $\ell_\infty$-norm. On an infinite collection of featured graphs, the topology is the disjoint union topology on $\bigsqcup_{\mathsf{G} \in \mathcal{G}} \mathbb{R}^{n_\mathsf{G} d}$, a compact subset consists of a finite set of graphs, and for each graph $\mathsf{G}$ only non-zero on a compact subset of $\mathbb{R}^{nd}$. For these reasons, we introduce Definition 5.1, which is still rich enough to distinguish between BuNNs and MPNNs.

# D WHY CLASSICAL ARGUMENTS DO NOT APPLY TO COMPACT UNIFORM APPROXIMATION

A seminal result in GNN expressivity is Theorem 2 in Morris et al. (2019). In this section, we discuss why the arguments do not hold for compact uniform approximation, even when enriched with injective positional encodings. We start by formally defining injective positional encodings.

**Definition D.1.** A positional encoding $\pi$ on a graph $\mathsf{G}$ is a map $\pi(\mathsf{G}) : \mathsf{V} \to \mathbb{R}^k$ which assigns a $k$-dimensional feature to every node . A positional encoding $\pi$ on a family of graphs $\mathcal{G}$ is a positional encoding on all graphs $\mathsf{G} \in \mathcal{G}$. A positional encoding $\pi$ is injective on $\mathcal{G}$ for any graph $\mathsf{G} \in \mathcal{G}$ and every node $u \in \mathsf{G}$ there is no other node $v \in \mathsf{H}$ for any $\mathsf{H} \in \mathcal{G}$ with $\pi(u) = \pi(v)$. In other words, each positional encoding corresponds to a unique node in the dataset.[4]

The setting in Morris et al. (2019) Theorem 2 considers a single graph with fixed node features (or colors/labels). In contrast, our Theorem is for a family of graphs, and more importantly, for each graph $\mathsf{G}$, the node features are not fixed but can vary in any compact subspace of feature space. This means that for a single graph of size $n$, our statement holds, for example, on the unit cube $[0, 1]^n$, while the result in Morris et al. (2019) only holds for a specific point in $\mathbb{R}^n$.

Fixing the node features is precisely what makes the construction in Morris et al. (2019) possible. Indeed the proof starts by assuming that the initial coloring is "linearly independent modulo equality", denoted by $\mathbf{F}_{l,0}^{(0)}$ in their proof. This property on node features is indeed central to the construction. It is used several times, for example "Observe that colors are represented by linearly independent row vectors" and "$\mathbf{F}_{l,0}^{(t+1)}$ is linearly independent modulo equality". Such an assumption is possible when dealing with fixed node features: since there are at most $n$ colors, it suffices to take a one-hot encoding of those colors, which fits in a feature space of dimension at most $n$. This assumption is also crucial for the injectivity of the sum aggregation.

In our setting, the node features can take any value in a continuum of features, each belonging to a compact subspace $K$ of $\mathbb{R}^c$. Encoding such a continuum as a one-hot encoding cannot be done in a finite-dimensional vector space (since there are $\{0, 1\}^K$ possibilities, which can be uncountable, and each need to be linearly independent). Hence, their construction fails in the setting where the node features are not fixed but can vary on any compact subspace of $\mathbb{R}^c$.

# E ALGORITHMIC AND IMPLEMENTATION DETAILS

In this section, we provide more details on the implementation of BuNNs. We start by discussing how to use several vector-field channels when the input dimension is greater than the bundle dimension. We then discuss how to use several bundles at once when a single bundle is insufficient. We then combine both views, namely having several vector-field channels on several bundles at once. Finally, we describe how we compute our bundle maps in the experiments.

**Extending to several vector-field channels**. When the signal dimension exceeds the bundle dimension, i.e. $c > d$, we cannot directly apply BuNNs to the input signal. In that case, we first transform the signal into a hidden dimension, a multiple of the bundle dimension, i.e. $c = dp$. We reshape the input signal into $p$ channels of $d$-dimensional vector fields, where we apply the diffusion step (Equation 3) on each $p$ channels simultaneously, and we apply the weight matrix $\mathbf{W} \in \mathbb{R}^{dp \times dp}$ by first flattening the node signals into $dp$ dimensions, then multiplying by $\mathbf{W}$, and then reshaping it into $p$ channels of $d$ dimensional vector fields.

**Extending to several bundles**. Learning a high dimensional orthogonal matrix $\mathrm{O}(d)$ becomes expensive since the manifold of orthogonal matrices is $\frac{d(d-1)}{2}$ dimensional. However, we can compute many low-dimensional bundles in parallel. In practice, we found that using several 2-dimensional bundles was enough. Computing $b$ different 2-dimensional bundles requires only $b$-parameters since the manifold $\mathrm{O}(2)$ is 1-dimensional. We, therefore, also use different 'bundle channels' given by an additional hyper-parameter – the number of bundles, which we denote $b$. Given an input signal of

---

[4]Note that on symmetric graphs such as the cycle, it is not possible to have injectivity if the positional encoding is equivariant. However, in practice many steps of the 1-WL color refinement will be equivariant and satisfy such assumption on most graphs.

dimension $c = db$, we can decompose the signal into $b$ bundle channels of dimension $d$. We can compute the diffusion step (Equation 3) for each bundle in parallel. For the update step (Equation 2), we apply the weight matrix $\mathbf{W} \in \mathbb{R}^{bd \times bd}$ by first flattening the node signals into $bd$ dimensions, then multiplying by $\mathbf{W}$, and then reshaping it into $b$ bundle channels of $d$ dimensional vector fields over $b$ different bundle structures.

*Remark* E.1. We note that using $b$ different $d$ dimensional bundles is equivalent to parameterizing a subset of one $bd$-dimensional structure, consisting of the orthogonal map $\mathbf{O} \in O(bd) \subset \mathbb{R}^{bd \times bd}$ that are block diagonal matrices $\mathbf{O} = \bigoplus_{i=1\ldots b} \mathbf{O}_i$, with each $\mathbf{O}_i \in O(d)$.

**Extending to several bundles and vector-field channels**. We can combine the above two observations. Given an input signal of dimension $c = bdp$, we can subdivide this into $b$ different bundle structures of dimension $d$ and $p$ channels for each bundle. We diffuse on the appropriate bundle structure and flatten the vector fields into a $c \times c$ vector before applying the learnable parameters.

**Computing the bundle maps**. In our experiments, we noticed that having several bundles of dimension 2 was more efficient than one bundle of large dimensions, while there was no clear performance gain when using higher dimensional bundles. To compute the $b$ bundle maps $\mathbf{O}_v$ we therefore only need $b$ rotation angles $\theta_v$, one per bundle. In our experiments, we use Housholder reflections using the python package Obukhov (2021) or direct parameterization. For direct parameterization, we do the following: since the matrix group $O(2)$ is disconnected, we always take $b$ to be even and parameterize half the bundles as rotation matrices $r(\theta) = \begin{pmatrix} \cos(\theta) & \sin(\theta) \\ -\sin(\theta) & \cos(\theta) \end{pmatrix}$ and the other half to correspond to matrices with determinant $-1$, which can be parameterized by $r^*(\theta) = \begin{pmatrix} \cos(\theta) & \sin(\theta) \\ \sin(\theta) & -\cos(\theta) \end{pmatrix}$. We compute the angles $\theta$ as in Equation 1 where the network $\phi^{(\ell)}$ is either an MLP or a GNN. The network $\phi$ is either shared across layers or differing at every layer.

**Taylor approximation algorithm**. We now provide pseudo-code on how we implement Equations 2, and 3. We then proceed with a complexity analysis. The key idea of the algorithm is that the bundle heat kernel can be approximated efficiently using the standard graph heat kernel.

---

**Algorithm 1 Taylor expansion implementation of a BuNN layer**

---

**Input**: Normalized graph Laplacian $\mathcal{L}$, Orthogonal maps $\mathbf{O}_v^{(\ell)} \ \forall v \in \mathsf{G}$, Node features $\mathbf{X}^{(\ell)} \in \mathbb{R}^{n \times d}$, Time $t$, Maximum degree $K$, Channel mixing matrix $\mathbf{W}^{(\ell)}$, bias $\mathbf{b}^{(\ell)}$
**Output**: Updated node features $\mathbf{X}^{(\ell)}$
1: $\mathbf{h}_v^{(\ell)} \leftarrow \mathbf{O}_v^{(\ell)} \mathbf{x}_v^{(\ell)} \ \forall v \in \mathsf{V}$        ▷ **Sync.:** Go to global representation
2: $\mathbf{H}^{(\ell)} \leftarrow \mathbf{H}\mathbf{W}^{(\ell)} + \mathbf{b}^{(\ell)}$        ▷ Update features with parameters
3: $\mathbf{X}^{(\ell+1)} \leftarrow \mathbf{H}^{(\ell)}$        ▷ approximation of degree 0
4: **for** $k = 1, \ldots K$ **do**
5:      $\mathbf{H}^{(\ell)} \leftarrow -\frac{t}{k}\mathcal{L}\mathbf{H}^{(\ell)}$        ▷ Term of degree $k$
6:      $\mathbf{X}^{(\ell+1)} \leftarrow \mathbf{X}^{(\ell+1)} + \mathbf{H}^{(\ell)}$        ▷ Approximation of degree $k$
7: **end for**
8: $\mathbf{x}_v^{(\ell+1)} \leftarrow \mathbf{O}_v^{(\ell)T} \mathbf{x}_v^{(\ell+1)} \ \forall v \in \mathsf{V}$        ▷ **Deync.:** Return to local representation
9: **return** $\mathbf{X}^{(\ell+1)}$

---

The complexity of the algorithms is as follows. There are 3 matrix-vector multiplications done at each node in lines 1, 2, and 8, which are done in $\mathcal{O}\left(3d^2|\mathsf{V}|\right)$. The for loops consist of matrix-matrix multiplications, which are done in $\mathcal{O}\left(d|\mathsf{E}|\right)$ with sparse matrix-vector multiplication. The memory complexity is $\mathcal{O}\left((d + d^2)|\mathsf{V}|\right)$ since we need to store $d$ dimensional vectors and the orthogonal maps for each node. The exact implementation is described in Algorithm 1

**Spectral method**. We now describe how to implement a BuNN layer using the eigenvectors and eigenvalues of the Laplacian.

---

**Algorithm 2 Spectral implementation of a BuNN layer**

---

**Input**: Eigenvectors and eigenvalues graph Laplacian $(\phi_i, \lambda_i)_i$, Orthogonal maps $\mathbf{O}_v^{(\ell)} \ \forall v \in \mathsf{G}$, Node features $\mathbf{X}^{(\ell)} \in \mathbb{R}^{n \times d}$, Time $t$, Maximum degree $K$, Channel mixing matrix $\mathbf{W}^{(\ell)}$, bias $\mathbf{b}^{(\ell)}$

**Output**: Updated node features $\mathbf{X}^{(\ell)}$

1: $\mathbf{h}_v^{(\ell)} \leftarrow \mathbf{O}_v^{(\ell)} \mathbf{x}_v^{(\ell)} \ \forall v \in \mathsf{V}$          ▷ **Sync.**: Go to global representation
2: $\mathbf{H}^{(\ell)} \leftarrow \mathbf{H}\mathbf{W}^{(\ell)} + \mathbf{b}^{(\ell)}$          ▷ Update features with parameters
3: $\mathbf{X}^{(\ell+1)} \leftarrow \sum_i e^{-t\lambda_i} \phi_i \phi_i^T \mathbf{H}^{(\ell)}$      ▷ Spectral solution to heat equation
4: $\mathbf{x}_v^{(\ell+1)} \leftarrow {\mathbf{O}_v^{(\ell)}}^T \mathbf{x}_v^{(\ell+1)} \ \forall v \in \mathsf{V}$     ▷ **Desync.**: Return to local representation
5: **return** $\mathbf{X}^{(\ell+1)}$

---

### E.1 HOUSEHOLDER REFLECTIONS.

Many different parameterizations of the group $O(n)$ exist. While direct parameterizations are possible for $n = 2, 3$ it becomes increasingly complex to do so for larger $n$, and a general method working for all $n$ is a desirata. While there are several methods to do so, we use Householder reflection since it is used in related methods (Bodnar et al., 2022). We use the Pytorch package from (**?**). Given given $k$ vectors $v_i \in \mathbb{R}^d$, define the Householder matrices as $H_i = I - 2\frac{v_i v_i^T}{\|v_i\|_2^2}$, and define $U = \prod_{i=1}^k H_i$. All orthogonal matrices may be obtained using the product of $d$ such matrices. Hence the map $\mathbf{R}^{d \times d} \to O(d)$ mapping $V = (v_i)$ to $U$ is a parameterization of the orthogonal group. We use pytorch implementations allowing autograd provided in (**?**).

## F EXPERIMENT DETAILS

In this section we provide additional information about the experiments on the heterophilic graph benchmarks, the LRGB benchmarks, and the synthetic experiments. All experiments were ran on a cluster using NVIDIA A10 (24 GB) GPUs, each experiment using at most 1 GPU. Each machine in the cluster has 64 cores of Intel(R) Xeon(R) Gold 6326 CPU at 2.90GHz, and ∼500GB of RAM available. The synthetic experiments from Section 6.1 were run on CPU and each run took roughly 20 minutes. The heterophilic experiments from Section 6 were run GPU and varied between 5 minutes to 1.5 hours. The LRGB experiments were run on GPU and varied between 0.5 hours and 4 hours.

### F.1 LRGB: TRAINING AND TUNING.

For `peptides-func` and `peptides-struct` we use a fixed parameter budget of $\sim 500k$ as in Dwivedi et al. (2022). We fix hyper-parameters to be the best GCN hyper-parameters from Tönshoff et al. (2023), and tune only BuNN-specific parameters as well as the use of BatchNorm. In Table 5, we report the grid of hyper-parameters that we searched, and denote in bold the best combinations of hyper-parameters. The parameters fixed from Tönshoff et al. (2023) are the following:

- Dropout 0.1
- Head depth 3
- Positional Encoding: LapPE for `struct` and RWSE for `func`
- Optimizer: AdamW with a cosine annealing learning rate schedule and linear warmup.
- Batch size 200
- We use skip connection as implemented in Dwivedi et al. (2022) and not in Tönshoff et al. (2023). That is, the skip connection does not skip the non-linearity.

For the BuNN specific parameters, we use 2 dimensional bundles, whose angles $\theta$ we compute with the help of a small SumGNN architecture using a sum aggregation as defined by $\theta_v^{(\ell)} = \sigma\left(\mathbf{W}_s \mathbf{x}_v^{(\ell)} + \mathbf{W}_n \sum_{u \in \mathcal{N}(v)} \mathbf{x}_u^{(\ell)}\right)$ where the input dimension is the hidden dimension, the hidden

dimension is twice the number of bundles and the output is the number of bundles. The number of SumGNN layers is a hyper-parameter we tune. When it is 0 we use a 2 layer MLP with hidden dimension also twice number of bundles. For each hyper-parameter configuration, we set the hidden dimension in order to respect to the parameter budget.

| Parameters | All Values | Best Values | | |
|---|---|---|---|---|
| | | func | struct | PascalVOC-SP |
| Num bundles b | $4, 8, 16$ | 16 | 16 | 16 |
| Number of BuNN layers | $1 - 20$ | 6 | 4 | 20 |
| Number of SumGNN layer | $0 - 3$ | 1 | 0 | 0 |
| Weight decay | $0, 0.1, 0.2, 0.3$ | 0.2 | 0.2 | 0.2 |
| Time $t$ | $0.1, 1, 10, 100$ | 1 | 1 | 1 |

Table 5: Grid of hyper-parameters for `peptides-func`, `peptides-struct`, and `PascalVOC-SP`.

### F.2 HETEROPHILIC GRAPHS: TRAINING AND TUNING.

For the heterophilic graphs we use the source code from Platonov et al. (2023) in which we add our layer definition. We report all training parameters that we have tuned. Namely, we use GELU activation functions, the Adam optimizer with learning rate $3 \times 10^{-5}$, and train all models for 2000 epochs and select the best epoch based on the validation set. To compute the bundle maps, we compute the parameters $\theta$ with a GraphSAGE architecture shared across layers ($\phi$ method = shared) or different at each layer ($\phi$ method = not shared), with hidden dimension dimension the number of bundles. The number of layers of this GNN is a hyper-parameter we tuned, which when set to 0 we use a 2 layer MLP. For each task we manually tuned parameters, which are subsets of the combinations of parameters in the grid from Table 6. The implementation of the heat kernel used is either truncated Taylor series with degree 8, or the spectral implementation. We report the best performing combination of parameters in Table 7. For the NSD baseline we use code from Bodnar et al. (2022) and tune equivalent parameters. We report the grid of hyperparameters in Table 8 and best values in Table 9

| Parameters | All Values |
|---|---|
| Hidden dim | $256, 512$ |
| Num bundles $b$ | $2, 4, 8, 16, 32, 64, 128, 256$ |
| Bundle dimension $d$ | $2$ |
| Number of BuNN layers | $1 - 8$ |
| Number of GNN layer $\phi$ | $0 - 8$ |
| Time $t$ | $0.1, 1, 10, 100$ |
| shared $\phi$ | ✗,✓ |
| Dropout | $0.0, 0.2$ |
| Learning rate | $3 \times 10^{-4}, 3 \times 10^{-5}$ |

Table 6: Parameters searched when tuning on the heterophilic graph benchmark datasets.

| Parameters | Best Values | | | | |
|---|---|---|---|---|---|
| | roman-empire | amazon-ratings | minesweeper | tolokers | questions |
| Hidden dim | 512 | 512 | 512 | 512 | 256 |
| Num bundles b | 32 | 4 | 128 | 256 | 1 |
| Bundle dim | 2 | 2 | 2 | 2 | 2 |
| Number of BuNN layers | 6 | 2 | 8 | 6 | 6 |
| Number of GNN layer | 8 | 0 | 8 | 7 | 6 |
| Time $t$ | 100 | 1.5 | 1 | 1 | 1 |
| shared $\phi$ | ✗ | ✗ | ✗ | ✗ | ✓ |
| Dropout | 0.2 | 0.2 | 0.2 | 0.2 | 0.2 |
| Learning rate | $3 \times 10^{-4}$ | $3 \times 10^{-4}$ | $3 \times 10^{-5}$ | $3 \times 10^{-5}$ | $3 \times 10^{-5}$ |

Table 7: Best parameter for each dataset in the heterophilic graph benchmarks.

| Parameters | All Values |
|---|---|
| Hidden dim | $64, 128, 256^*, 512^*$ |
| Sheaf dimension $d$ | 2, 4, 8 |
| Number of layers | $1, 2, 3, 4^*, 5^*, 6^*, 7^*, 8^*$ |
| Dropout | 0.0, 0.2 |
| Learning rate | $3 \times 10^{-4}, 3 \times 10^{-5}$ |

Table 8: Parameters searched when tuning NSD on the heterophilic graph benchmark datasets. Parameters marked by $^*$ ran out of memory on some datasets.

| Parameters | Best Values | | | | |
|---|---|---|---|---|---|
| | roman-empire | amazon-ratings | minesweeper | tolokers | questions |
| Hidden dim | 256 | 256 | 512 | 64 | 64 |
| Sheaf dim | 2 | 4 | 2 | 2 | 2 |
| Number of layers | 8 | 3 | 8 | 5 | 5 |
| Dropout | 0.2 | 0.2 | 0.2 | 0.2 | 0.2 |
| Learning rate | $3 \times 10^{-4}$ | $3 \times 10^{-4}$ | $3 \times 10^{-4}$ | $3 \times 10^{-4}$ | $3 \times 10^{-4}$ |

Table 9: Best parameter for NSD on each dataset in the heterophilic graph benchmarks.

## G  EMPIRICAL RUNTIME

Table 10: Average training time, over 3 runs, for different architectures of with 5 layers and hidden dimension of 512 on the Heterophilic datasets (averaged over 3 runs). BuNN and NSD use a single 2 dimensional sheaf. BuNN is implemented spatially. All experiments were performed on an NVIDIA A10 (24GB) GPU.

| *avg. training time* | roman-empire | amazon-ratings | minesweeper | tolokers | questions |
|---|---|---|---|---|---|
| # num nodes | 22,662 | 24,492 | 10,000 | 11,758 | 48,921 |
| # num edges | 32,927 | 93,050 | 39,402 | 519,000 | 153,540 |
| SAGE | 1:45 | 1:43 | 0:48 | 1:15 | 3:01 |
| GAT | 2:33 | 2:42 | 1:11 | 3:24 | 5:22 |
| GT | 3:31 | 4:12 | 1:52 | 4:20 | 7:57 |
| NSD | 7:58 | 9:16 | 7:13 | OOM | OOM |
| BuNN | 4:21 | 5:18 | 2:48 | 3:51 | 9:32 |

## H  ABLATION: POSITIONAL ENCODING ABLATION

We perform an ablation on the use of PE on the `peptides-func` and `peptides-struct` datasets. We retrain a model using the best hyperparameter where we change the used PEs. We compare the two main PE used in graph machin learning, namely Laplacian Positional encoding (LPE) and Random Walk Structural Encodings (RWSE), and we consider using no PE as a baseline. Results can be found in Table 12. We observe that using PE is always beneficial to not using PE, however each task seem to admit a preferred PE, since LPE is better for `Peptides-struct` and RWSE for `Peptides-func`.

## I  ABLATION: IMPORTANCE OF $W$ AND $b$

We run an ablation on the importance of the $\mathbf{W}$ and $b$ parameters in Equation 2. We retrain a model removing them and compare it to a model trained using them. We report results in Table 13. The results suggest that these parameters help. However, the result suggest that they are not essential to achieve good results, as even without them, the model performs well and beats all but one of the baselines on `peptides-func`.

Table 11: Average time per epoch, over 5 epochs, for different architectures with 6 layers and 500k parameters on the LRGB datasets. BuNN is implemented spectrally and uses a single 2 dimensional bundle. All experiments were performed on an NVIDIA A10 (24GB) GPU.

| *avg. time / epoch* | Peptides-func | Peptides-struct | PascalVOC-SP |
|---|---|---|---|
| # graphs | 15,535 | 15,535 | 11,355 |
| # nodes | 2,344,859 | 2,344,859 | 15,955,687 |
| # edges | 4,773,974 | 4,773,974 | 32,341,644 |
| GCN | 5.6s | 5.3s | 15.2s |
| GatedGCN | 8.5s | 8.3s | 22.5s |
| GPS | 17.2s | 17.2s | 38.2s |
| BuNN | 12.7s | 12.5s | 28.7s |

Table 12: Positional encoding ablation on peptides-func and peptides-struct

| | RWSE | LPE | No PE |
|---|---|---|---|
| Peptides-func **Test AP** $\uparrow$ | $\mathbf{72.76 \pm 0.65}$ | $72.25 \pm 0.51$ | $71.76 \pm 0.68$ |
| Peptides-struct **Test MAE** $\downarrow$ | $25.02 \pm 0.15$ | $\mathbf{24.63 \pm 0.12}$ | $25.32 \pm 0.19$ |

## J   TREE-NEIGHBORSMATCH TASK

As an additional synthetic task, we evaluate BuNNs on the `Tree-NeighborsMatch` task, proposed in Alon & Yahav (2021) to show that MPNNs suffer from over-squashing. We use their code and setup to evaluate the capacity fo BuNNs to alleviate over-squashing. We use their reported results and add our own results, ran using their experimental setup, for a 2-layer deep BuNN with $t = \infty$ and 32 bundles. We report the results in Figure 5. We observe that BuNN beat all MPNNs by a large margin, with perfect until $r = 6$. As the task gets harder with a larger depth, the accuracy for BuNNs drops slower than the accuracy of MPNNs. These results confirm that BuNNs alleviate over-squashing.

Table 13: $W$ and $b$ importance ablation on `peptides-func` and `peptides-struct`

| | with | without |
|---|---|---|
| Peptides-func **Test AP** $\uparrow$ | $\mathbf{72.76 \pm 0.65}$ | $70.75 \pm 0.36$ |
| Peptides-struct **Test MAE** $\downarrow$ | $\mathbf{24.63 \pm 0.12}$ | $25.28 \pm 0.32$ |

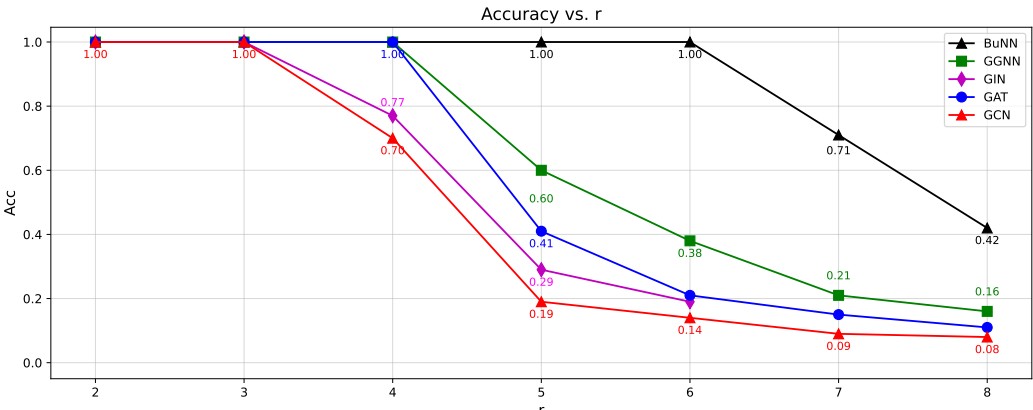

Figure 5: Results for `Tree-NeighborsMatch` task.

