# OpenReview forum: "Bundle Neural Network for message diffusion on graphs"
_ICLR.cc/2025/Conference — ICLR 2025 Spotlight_

### Official Review · Reviewer_uDEq · 2024-10-31

**Soundness:** 3
**Presentation:** 2
**Contribution:** 2
**Rating:** 8
**Confidence:** 4

**Summary:**

This paper proposes to perform message-passing over a diffused graph using flat vector bundles. This is achieved by transforming each node state within its own vector space using a learned and distinct orthogonal map. This is the message that a node sends to each neighbor, where the inverse of the orthongonal map from the receiving node is used as a decoding mapping. The authors show that this can avoid over-smoothing, over-squashing, and can be a compact uniform approximation over a set of graphs. Experiments show the effectiveness of the proposed approach.

**Strengths:**

The proposed method is sound and very general. BuNNs have many favorable and interesting properties. I particularly like the property regarding fixed points. The paper is nicely written and mostly understandable.

**Weaknesses:**

* Eq. 2: From the theoretical derivation, this step does not seem needed as $\mathcal{H}_B$ already applies $O_v$ and $O_u$. The only provided reason is that it helps to prove Theorem 5.3. Are there other reasons why $W$ and $b$ are introduced? What happens when these are removed?
* BuNNs and their properties are mostly compared to MPNNs applied to the original. It seems fairer to me to also compare its properties to Graph Transformers and infinite-depth MPNNs (e.g. [1]) (Table 1).
* Theorem 5.3:
    * This statement is quite interesting but I do not fully understand it. In which cases are injective positional encodings achievable? Would it be satisfied when $\mathcal{G}$ is a set of WL distinguishable graphs and we use WL colors are positional encodings?
    * How does it deal with two nodes $u,v$ in the same graph with the same WL unfolding tree? By my understanding of the injective positional encodings, they could still have the same initial state. The MLP (Eq. 1) would produce the same $\mathbf{O}_v = \mathbf{O}_u$ for both nodes.
    * Which changes would be needed for graph transformers to also satisfy this universality?
* Table 2: A comparison with Graph Transformers or infinite-depth MPNNs would be insightful
* Table 3: It seems like the authors performed experiments for NSD and BuNN, but hyperparameter ranges and optimal hyperparameters are only presented for BuNN.
* Details about runtimes are missing. It would be important to have execution times for at least one experiment for BuNN for a comprehensive evaluation.

---
[1] Gu et al., Implicit graph neural networks, NeurIPS (2020).

**Questions:**

* See weaknesses

---

> ### Author Response · Authors · 2024-11-19
>
> We thank the reviewer for their time and effort spent reviewing our work. We are pleased to hear that the reviewer found the proposed method “sound and general”, that BuNNs have “many favourable and interesting properties”, and that our paper is “nicely written”.
>
> We now address the questions raised by the reviewer:
>
> > *Are there other reasons why $\mathbf{W}$ and $b$ are introduced? What happens when these are removed?*
>
> We originally incorporated $\mathbf{W}$ and $b$ for consistency with the previous work of Bodnar et al. [3]. Additionally, as mentioned in our work, these play an important role in proving Theorem 5.3. That said, the reviewer questions whether these parameters actually matter in practice, since Equation $2$ already incorporates the learned restriction maps. In order to test this, we performed an additional ablation on the peptides-func and peptides-struct datasets. We report details and results in Table 13 of Appendix I. The results suggest that these additional parameters help achieve better performance, but even without them, BuNN outperforms all but one baseline on the peptides-func dataset.
>
> > *BuNNs and their properties are mostly compared to MPNNs applied to the original. It seems fairer to me to also compare its properties to Graph Transformers and infinite-depth MPNNs (e.g. [1]) (Table 1).*
>
> We agree with the reviewer that discussing the properties of other paradigms is interesting. However, we would like to emphasize that our work explores how BuNNs and message diffusion can address the main limitations of MPNNs, rather than explicitly comparing them to all other architectures. That said, we have followed the reviewer’s suggestion and added a discussion on Graph Transformer and Implicit Graph Neural Networks [1] in the main text Section 3.
>
> > *Table 2: A comparison with Graph Transformers or infinite-depth MPNNs would be insightful*
>
> We respectfully reiterate our previous point, our work studies how BuNNs and message diffusion can address the main limitations of MPNNs, rather than comparing them to all other paradigms. For this reason, we had not included other paradigms in the synthetic experiment on over-smoothing and over-squashing since these have been studied mostly for MPNNs. That said, for the sake of completeness, we follow the reviewers advice and have added a fully connected Graph Transformer (GraphGPS, [2]) as a baseline in Table 2. The results were surprising; while the model achieves good training performance, it is overfitting and performs comparably to MPNNs on the test set. We believe that this is an artifact of the relatively small training set, as Transformers are known to be very data-hungry.

---

> > ### Author Response · Authors · 2024-11-19
> >
> > > *Theorem 5.3:
> > This statement is quite interesting but I do not fully understand it. In which cases are injective positional encodings achievable? Would it be satisfied when  is a set of WL distinguishable graphs and we use WL colors are positional encodings?*
> >
> > The reviewer is correct: in the case where the dataset consists of WL-distinguishable graphs, the WL colors will consist of injective positional encodings. Another powerful encoding are using the eigenvectors of the Laplacian, which are injective when the graphs do not contain repeated eigenvalues. However, injective positional encodings are not always achievable.
> >
> > > *How does it deal with two nodes  in the same graph with the same WL unfolding tree? By my understanding of the injective positional encodings, they could still have the same initial state. The MLP (Eq. 1) would produce the same  for both nodes.*
> >
> > The reviewer is correct: if two nodes have the same WL-coloring, and the positional encoding is based on the WL test (or a GNN), then the assumptions of Theorem 5.3. do not hold. However, in this case, it might be possible that another PE is injective on these nodes.
> >
> > > *Which changes would be needed for graph transformers to also satisfy this universality?*
> >
> > This is an open problem, and understanding if other architectures such as Graph Transformers satisfy the universality property is an exciting direction, albeit outside the scope of our work.
> >
> > > *Table 3: It seems like the authors performed experiments for NSD and BuNN, but hyperparameter ranges and optimal hyperparameters are only presented for BuNN.*
> >
> > We thank the reviewer for pointing this out. We have reported the hyperparameter range and optimal hyperparameters for NSD in Tables 8 and 9 of Appendix F of the updated paper.
> >
> > > *Details about runtimes are missing. It would be important to have execution times for at least one experiment for BuNN for a comprehensive evaluation.*
> >
> > We thank the reviewer for pointing this out. We have added the runtimes on all datasets and baselines in Tables 10 and 11 of Appendix G of the updated paper. On larger graphs (PascalVOC) BuNNs is about 30% faster than GPS.
> >
> >
> >
> > Again, we thank the reviewer for their time and effort in reviewing our work. We appreciate the insightful comments and hope to have addressed them in full. If so, we kindly ask you to reevaluate our work from a fresh perspective and potentially consider a score upgrade. If not, we look forward to answer any further question you may have.
> >
> >
> > ### References:
> >
> > [1] Gu, Fangda, et al. "Implicit graph neural networks." Advances in Neural Information Processing Systems 33 (2020): 11984-11995.
> >
> > [2] Rampášek, Ladislav, et al. "Recipe for a general, powerful, scalable graph transformer." Advances in Neural Information Processing Systems 35 (2022): 14501-14515.

---

> > > ### Comment · Reviewer_uDEq · 2024-11-24
> > >
> > > I thank the authors for their dedicated rebuttal and clarifications. My concerns have been addressed, and I would like to see this paper accepted. I raised my score accordingly.
> > >
> > > I would like to see one final detail clarified regarding this response:
> > > > The reviewer is correct: if two nodes have the same WL-coloring, and the positional encoding is based on the WL test (or a GNN), then the assumptions of Theorem 5.3. do not hold. However, in this case, it might be possible that another PE is injective on these nodes.
> > >
> > > My understanding is that the injective positional encoding still allows two nodes in the same graph to get assigned the same positional encoding. Let's say two nodes in the same graph are assigned the same encoding, and those nodes cannot be further distinguished by WL. Then, BuNN would always assign them the same representation. By your definition of a feature transformation, there should exist one that assigns these two nodes to different representations. How does BuNN deal with such a particular case? If there are errors in my understanding, please correct me.

---

> > > > ### Author Response · Authors · 2024-11-24
> > > >
> > > > We are happy to hear that we have addressed the reviewer’s concerns, and we thank the reviewer for reevaluating our work.
> > > >
> > > > We now answer the interesting and subtle question asked by the reviewer. In the case of an injective positional encoding, two nodes can be assigned the same positional encoding only if they are isomorphic (i.e. there is an isomorphism between the two graphs that maps one node to the other). In this case, since the nodes are isomorphic, no isomorphism test could ever distinguish them, and therefore it is natural for the two nodes to result in the same representation. That said, the input features (which are different from the positional encodings) can still be different for the two nodes, and this could result in different output representations for these two nodes. We hope that this clarifies the question, and we are happy to further discuss this subtle point.
> > > >
> > > > We thank you for engaging in the rebuttal, and for your time and valuable feedback.

---

### Official Review · Reviewer_rcKT · 2024-11-02

**Soundness:** 2
**Presentation:** 2
**Contribution:** 2
**Rating:** 6
**Confidence:** 2

**Summary:**

The paper proposes a new graph neural network architecture called Bundle Neural Networks (BuNNs). The architecture is based on message diffusion on flat vector bundles which are topological structures that assign to each node a vector space and an orthogonal map. They seem to be inspired from Sheaf neural networks and, in different forms, are claimed to address the problems of overs-moothing and over-squashing. Theoretical analysis on the feature transformation with injective positional encodings shows uniform expressivity. Further analysis on the properties of BuNNs and experimental results on both synthetic and real-world datasets, are provided for validating the proposed model.

**Strengths:**

1. The paper addresses an important problem of oversmoothing and oversquashing seen in GNNs via vector bundles, building on cellular sheafs in GNNs.
1. The use of flat vector bundles with orthogonal maps to reduce the computational complexity is interesting.
1. The paper is well-organized.

**Weaknesses:**

1. The experiments to validate oversmoothing and oversquashing are not comprehensive. The authors can test the oversquashing on Neighbours match dataset [1], which has been used well in multiple papers. Further the authors can explore using effective resistance metric to measure oversquashing [3] and [4].
1. Related works which are designed via dirichlet energy optimization like [2] are not discussed. May help to discuss the difference with [2] in terms of results on oversmoothing.

### References:
[1]$~$ Uri Alon and Eran Yahav. On the bottleneck of graph neural networks and its practical implications. arXiv preprint arXiv:2006.05205, 2020.

[2]$~$ Fu, Guoji, et al. "Implicit graph neural diffusion based on constrained Dirichlet energy minimization." arXiv preprint arXiv:2308.03306 (2023).

[3]$~$ Mitchell Black, Zhengchao Wan, Amir Nayyeri, and Yusu Wang. Understanding oversquashing in gnns through the lens of effective resistance. In International Conference on Machine Learning, pages 2528–2547. PMLR, 2023.

[4]$~$ Dong, Yanfei, et al. "Differentiable Cluster Graph Neural Network." arXiv preprint arXiv:2405.16185 (2024).

**Questions:**

Please see weaknesses

---

> ### Author Response · Authors · 2024-11-19
>
> We thank the reviewer for their feedback and for finding our paper “well organized”, that it “addresses important topics”, and the reduced computational complexity of BuNNs over Sheaves “interesting”.
>
> We provide detailed responses to the reviewer’s comments below:
>
> > *The authors can test the oversquashing on Neighbours match dataset [1]*
>
> We follow the reviewer’s suggestion and test the capacity of BuNNs to mitigate over-squashing on the Tree-NeighborsMatch dataset from [1]. We use the experimental setup and code form [1] and report our results in Figure 5 in Appendix J of the updated paper. BuNNs outperform all MPNNs for all values of r, getting perfect accuracy up to r=6, which is over twice as good as the best MPNN. This confirms our claim that BuNNs do alleviate over-squashing.
>
> > * Further the authors can explore using effective resistance metric to measure oversquashing [3] and [4].*
>
> While we mention the barbell graphs’ high commute time between nodes from opposite clusters, we do not explicitly mention the effective resistance. We thank the reviewer for suggesting these important references. We have updated the paper accordingly in Section 6.1 and have added both citations.
>
> > *Related works which are designed via dirichlet energy optimization like [2] are not discussed. May help to discuss the difference with [2] in terms of results on oversmoothing.*
>
> We thank the reviewer for the suggestion. We added a discussion of related results, including [2], in the main text Section 4.1.
>
> We thank the reviewer for their valuable feedback. We hope to have addressed the weaknesses, and we are happy to discuss them further. If not, we kindly ask the reviewer to consider reevaluating our work from a fresh perspective and to consider a score increase potentially.
>
> ## References:
>
> [1]$~$ Uri Alon and Eran Yahav. On the bottleneck of graph neural networks and its practical implications. arXiv preprint arXiv:2006.05205, 2020.
>
> [2]$~$ Fu, Guoji, et al. "Implicit graph neural diffusion based on constrained Dirichlet energy minimization." arXiv preprint arXiv:2308.03306 (2023).
>
> [3]$~$ Mitchell Black, Zhengchao Wan, Amir Nayyeri, and Yusu Wang. Understanding oversquashing in gnns through the lens of effective resistance. In International Conference on Machine Learning, pages 2528–2547. PMLR, 2023.
>
> [4]$~$ Dong, Yanfei, et al. "Differentiable Cluster Graph Neural Network." arXiv preprint arXiv:2405.16185 (2024).

---

> ### Comment · Area_Chair_pJZS · 2024-11-26
>
> Please check if the authors' response addresses your concerns.

---

> ### Comment · Reviewer_rcKT · 2024-11-28
>
> Thank you for your response. I maintain my score and positive view on the paper.

---

> > ### Author Response · Authors · 2024-11-29
> >
> > Given the remaining time in the rebuttal, we politely ask the reviewer if there are other aspects of our work we could improve that would allow the reviewer to evaluate our work even more positively.
> >
> > We thank the reviewer for their time and valuable feedback and for engaging in the rebuttal process.

---

### Official Review · Reviewer_9bAu · 2024-11-03

**Soundness:** 3
**Presentation:** 2
**Contribution:** 3
**Rating:** 8
**Confidence:** 4

**Summary:**

This paper extends previous work on graph neural networks equipped with cellular sheaves by introducing a more efficient way of computing the heat diffusion over bundles. The key idea of the authors is to assume a simpler form for the sheaves, namely a flat vector bundle, which can be thought of as an orthogonal map associated with each node (as opposed to one associated with each node-edge pair). This assumption significantly simplifies the computations necessary to learn the bundles from data. In particular, the heat diffusion kernel can now be computed efficiently during the learning of the bundles. This enables the authors to use spectral methods to run the diffusion process for long periods of time addressing the over-squashing problem on graphs. This flat vector bundles also bring the strength of the cellular sheaves in that the fixed point of diffusion is no longer a trivial state where all nodes share the same features. Rather, the long time limit now has richer structure where variation is allowed across the nodes. This addresses the over-smoothing problem. The authors propose GNN architecture using flat vector bundles and heat kernels (computed both using Taylor expansions and spectral methods) and demonstrate improved performance on synthetic data sets and real data sets.

**Strengths:**

This reviewer generally liked the paper. The idea of simplifying cellular sheaves using flat vector bundles is clever. This idea seems to solve the key problem with the computational complexity challenges faced by neural sheaf diffusion. Another strength of the paper is the clever synthetic toy models which demonstrate the gain from the proposed bundle architecture very clearly, especially when it comes to the over-squashing problem. The empirical performance of the proposed method is also impressive, especially on heterophilic graphs like minesweeper. The novel notion of expressivity introduced by the authors is also intriguing.

**Weaknesses:**

A weakness of this paper is that the central idea is arguably a simple extension of the neural sheaf diffusion paper. It is not clear that this constitute significant advance. This is balanced by the impressive empirical results in the paper.

The authors highlight well the advantages of replacing the cellular sheaves with flat vector bundles. However, it is not clear what is the cost of doing so. How does expressivity suffer with this assumption? Are there circumstances under which this simplification works better? For example, does the sparsity of the graph play role. More discussion on this would have helped the reader. The authors nicely show the existence of fixed points for the diffusion process under flat vector bundles and that features across nodes are related by orthogonal transformations. How does this compare with the structure of fixed points under general cellular sheaves? How does this impact the expressivity of the bundle neural networks compared with their sheave counterpart?

This reviewer struggled at times to understand how much the principled geometric approach brought over from topology and differential geometry contributed to formulation of the proposed GNN. Although interesting, can the author's approach be though of as learning a transformation on each node. With the formality relaxed, it is not clear if this transformation needs to be orthogonal. The authors seems to go to quite a bit of trouble to ensure that the learned matrices at each node are orthogonal. Of course, Riemannian manifolds require this, but is it really needed? Would invertible matrices done just as good of a job? It seems like in practice they are limited to using only 2-dimensional vector spaces on each node, with the transformation matrices parameterized explicitly to be orthogonal.

This reviewer was bothered at times by the strong use of language by the authors. The claim that the proposed bundle approach solves the over-squashing problem is over-stated. Any spectral method will address over-squashing. The authors have proposed an efficient way to compute the heat kernel more efficiently by replacing cellular sheaves with flat vector bundles. This allows them to run diffusion for longer times. There is no conceptual new solution to over-squashing.

**Questions:**

What price is paid for going rom the full cellular sheaves to the flat vector bundle in expressivity? In practice, in what settings is the flat vector bundle approach good enough? For example, it seems like that for sparse graphs the orthogonal maps associated with each node would be good enough.

Similarly, can the authors provide a comparison between the structure of the fixed points for cellular sheaves and flat vector bundles? It seems like some of the "richness" of the long time limit of diffusion is lost with the flat vector bundles. How does this reflect on expressivity of flat vector bundles compared with cellular sheaves?

Does the transformation learned for each node need to be orthogonal in practice? It seems that the approach is somewhat limited because the learned matrices must be orthogonal. It is difficult to learn matrices that are constrained to be orthogonal. Can the authors relax this requirement in practice and use invertible matrices or a fully unconstrained matrix?

It seems like a key aspect of the proposed approach is how the orthogonal transformations at each node are learned? The authors mention MLPs or even GNNs that take in the graph structure and positional encodings of the nodes. What type of positional encodings are best for learning the orthogonal transformations? Why are the node features not taken into account when computing the orthogonal map for each node? Are the authors trying to keep the graph geometry distinct from the feature information?

In Table 3, how does the BuNN approach outperform the NSD for the data sets in which memory is not a limit for NSD? Is the depth of the models in BuNN larger helped by spectral methods? It seems like depth was 1 for the amazon-ratings data set. Does NSD overfit the data because of its use of cellular sheaves?

---

> ### Author Response · Authors · 2024-11-19
>
> We thank the reviewer for the feedback and positive comments. We are happy that the reviewer generally liked the paper and that they found the simplification of sheaves to flat bundles clever, the synthetic experiments clever, the empirical performance impressive, and the novel notion of expressiveness intriguing.
>
> We now address the questions and weaknesses raised:
>
> > *A weakness of this paper is that the central idea is arguably a simple extension of the neural sheaf diffusion paper. […] This is balanced by the impressive empirical results in the paper.*
>
>
> We agree that neural sheaf diffusion (NSD) and our work are similar in nature, but we respectfully disagree with the reviewer that the central idea of our paper is a simple extension of theirs. We explicitly state the differences with Sheaf Neural Networks in Section 3 of our paper, which we repeat here: 1) the NSD architecture is a message-passing architecture where each layer performs a one-hop aggregation, while in BuNNs the parameterization of the heat kernel breaks away from explicit message passing; 2) the use of flat vector bundles reduces the computational complexity; 3) the properties of the stable states of heat diffusion being non-trivial for flat vector bundles which is not true for general sheaves. In addition, we consider the theoretical properties of our method such as the universality result to be solid non-trivial contributions and are of independent interest.
>
> > *This reviewer was bothered at times by the strong use of language by the authors. The claim that the proposed bundle approach solves the over-squashing problem is over-stated. Any spectral method will address over-squashing. [...] There is no conceptual new solution to over-squashing.*
>
> We would like to clarify that we do not claim that our method is a new way to solve over-squashing and simply argue that our method does alleviate over-squashing. Alleviating over-squashing is a by-product of the proposed method, and we study this property as it is a main limitation of MPNNs. That said, we appreciate that the wording in Table 1 is strong, and we have replaced ‘No over-squashing’ with ‘Alleviates over-squashing.
>
> > *What price is paid for going from the full cellular sheaves to the flat vector bundle in expressivity? In practice, in what settings is the flat vector bundle approach good enough?*
>
> The reviewer raises an interesting point: since flat vector bundles are a special case of cellular sheaves, when is such a special case good enough, both in theory and in practice? While cellular sheaves are more general, our theory shows that flat vector bundles are sufficiently expressive to be **universal** (Theorem 5.3). In practice, NSD has to learn restriction maps for each node-edge pair, which is expensive and prone to overfitting. Our experiment strongly suggests that flat vector bundles are sufficiently expressive and work better in practice. We hope that this clarifies one of the messages of our paper.
>
> > *Similarly, can the authors provide a comparison between the structure of the fixed points for cellular sheaves and flat vector bundles?*
>
> The reviewer raises a second interesting point related to expressiveness: are the fixed points of flat vector bundles less ‘rich’ than those of general vector bundles? Concerning this, it is important to note that a key assumption of the previous work on the long time limit of general cellular sheaves by Bodnar et al. is that the bundle is path-independent, meaning that the transport map between two nodes does not depend on the path along which to transport. When this assumption does not hold, as is usually the case for a general sheaf, the fixed points are restricted to a strict subspace (c.f. Lemma 6 in [1]). We have updated Section 3 to clarify this comparison and are happy to discuss this further.

---

> > ### Author Response · Authors · 2024-11-19
> >
> > > *Does the transformation learned for each node need to be orthogonal in practice? [...] Can the authors relax this requirement in practice and use invertible matrices or a fully unconstrained matrix?*
> >
> > The reviewer makes a valid point, generalizing to invertible and unconstrained matrices is possible, and Bodnar et al. has investigated unconstrained matrices for cellular sheaves, but found that orthogonal sheaves generally work better in practice. Additionally, orthogonality is useful to guarantee the properties with respect to over-smoothing as the transpose is the inverse, and it means that their inverse is trivial to compute and numerically stable (for this reason, orthogonal matrices have long been popular also in recurrent neural networks). To conclude, we agree that in principle non-orthogonal maps could be used, however, we limit our attention to orthogonal maps due to the previous empirical evidence as well as computational efficiency and numerical stability consideration.
> >
> > > *It seems like a key aspect of the proposed approach is how the orthogonal transformations at each node are learned? What type of positional encodings are best for learning the orthogonal transformations?*
> >
> > The reviewer is correct, one difference between NSD and BuNN is that we allow for more flexibility to compute the restriction maps and adding positional encodings. To test what positional encodings were best for BuNNs, we perform an ablation study on the peptides-func and peptides-struct datasets, where we evaluated BuNN using the two most popular PE methods, namely Laplacian Eigenvectors (LPE) and Random Walk Structural Encodings (RWSE), and compre these to using no positional Encodings. Results can be found in Table 12 of Appendix H of the Appendix. The results show that the choice of PE is important and that using PE is always beneficial, however, the best PE depends on the dataset.
> >
> >
> > > *Why are the node features not taken into account when computing the orthogonal map for each node?*
> >
> > The reviewer is correct, the features can also be taken into into account to construct the restriction maps and in fact we do use them in our experiments. We thank you for pointing this out, we will clarify this point in the text and updated the Equation and the main Figure.
> >
> >
> > > *In Table 3, how does the BuNN approach outperform the NSD for the data sets in which memory is not a limit for NSD? Is the depth of the models in BuNN larger helped by spectral methods? It seems like depth was 1 for the amazon-ratings data set. Does NSD overfit the data because of its use of cellular sheaves?*
> >
> > Indeed, on the amazon-ratings dataset, it seems like NSD overfitts to the data. We believe that this is due to the fact that NSD learns a restriction map for every edge leading to a larger flexibility to BuNNs. As the reviewer points out, this is accentuated for amazon-ratings, which has a large number of edges, making NSD more prone to overfitting. This would suggest that the BuNNs acts as a better regularizer on graphs with a large number of edges compared to NSD which can overfit to the edges.  We have added a discussion of this is the updated text.
> >
> >
> > We thank the reviewer for their feedback, and for the many interesting points raised. We hope to have provided clarifications to the different questions raised. We are of course happy to further discuss any questions you may have.  If not, we kindly ask the reviewer to consider reevaluating our work from a fresh perspective and to consider a potential score upgrade.
> >
> >
> > [1] Bodnar, Cristian, et al. "Neural sheaf diffusion: A topological perspective on heterophily and oversmoothing in gnns." Advances in Neural Information Processing Systems 35 (2022): 18527-18541.

---

> > > ### Comment · Reviewer_9bAu · 2024-11-26
> > >
> > > The authors have addressed this reviewer's concerns through their replies and revisions. Accordingly, the score has been upgraded.

---

> > > > ### Author Response · Authors · 2024-11-29
> > > >
> > > > We thank the reviewer for their time, valuable feedback, and reevaluating our work.

---

> ### Comment · Area_Chair_pJZS · 2024-11-26
>
> Please check if the authors' response addresses your concerns.

---

### Author Response · Authors · 2024-11-19
**Global response**

We sincerely thank the reviewers for their valuable feedback and time. We are happy that the paper was generally well received. We are encouraged to see that reviewers found the idea of simplifying cellular sheaves using flat vector bundles “clever” (**9bAu**) and its reduced computational complexity “interesting” (**rcKT**). The method was noted for “addressing important topics” (**rcKT**) and having “many favorable and interesting properties” (**uDEq**). The reviewers appreciated the “clever” synthetic experiments (9bAu) and found the empirical results “impressive” (**9bAu**). The proposed notion of expressiveness was deemed “novel” and “intriguing” (**9bAu**), and Theorem 5.3 was described as “quite interesting” (**uDEq**). Lastly, we are pleased that the paper was considered “nicely written” (**uDEq**) and “well-organized” (**rcKT**).

We have uploaded an improved version of the paper incorporating the reviewers' comments and have colored in blue any modifications. Before addressing individual comments, we list the main improvements made to the paper:

- Added experiments on Tree-NeighborsMatch task to test the model’s capacity to alleviate over-squashing in Appendix J. (**rcKT**)
- Ablations on the importance of positional encodings (**9bAu**) and parameterization (**uDEq**) in Appendix H and I, respectively.
- Added empirical runtimes of different architectures on all datasets in Tables 10 and 11 in Appendix G. (**uDEq**)
- Added hyperparameter ranges and optimal parameters for NSD on heterophilic graphs in Tables 8 and 9 in Appendix F.2. (**uDEq**)

Again, we thank the reviewers for their valuable time and insightful feedback. We hope that with these modifications and the individual responses to the reviewers, we were able to address all questions. We look forward to further discussion.

---

### Meta-Review · Area_Chair_pJZS · 2024-12-19

**Metareview:**

The manuscript proposes a novel graphical neural network architecture that sends messages on flat vector bundles. The reviewers unaminously supports the manuscript for acceptance. During the discussion phase, the authors also address most of the concerns by the reviewers. Therefore, the metareviewer would recommend acceptance of the paper.

**Additional Comments On Reviewer Discussion:**

The authors have addressed the reviewers' concerns thoroughly through the discussion phase.

---

### Decision · Program_Chairs · 2025-01-22

Accept (Spotlight)